# PTH regulates osteogenesis and suppresses adipogenesis through Zfp467 in a feed-forward, PTH1R-cyclic AMP-dependent manner

Hanghang Liu[1,2], Akane Wada[3,4], Isabella Le[1,5], Phuong T Le[1], Andrew WF Lee[1,6], Jun Zhou[3,4], Francesca Gori[3], Roland Baron[3,4†], Clifford J Rosen[1*†]

[1]Maine Medical Center Research Institute, Maine Medical Center, Scarborough, United States; [2]West China Hospital of Stomatology, Sichuan University, Sichuan, China; [3]Division of Bone and Mineral Research, Dept of Oral Medicine, Infection and Immunity, Harvard School of Dental Medicine, Boston, United States; [4]Harvard Medical School, Department of Medicine and Endocrine Unit, Massachusetts General Hospital, Boston, United States; [5]Graduate Medical Sciences, Boston University School of Medicine, Boston, United States; [6]University of New England, College of Osteopathic Medicine, Biddeford, United States

*For correspondence:
cjrofen@gmail.com

†Co-senior authors

Competing interest: The authors declare that no competing interests exist.

**Abstract** Conditional deletion of the PTH1R in mesenchymal progenitors reduces osteoblast differentiation, enhances marrow adipogenesis, and increases zinc finger protein 467 (*Zfp467*) expression. In contrast, genetic loss of *Zfp467* increased *Pth1r* expression and shifts mesenchymal progenitor cell fate toward osteogenesis and higher bone mass. PTH1R and ZFP467 could constitute a feedback loop that facilitates PTH-induced osteogenesis and that conditional deletion of *Zfp467* in osteogenic precursors would lead to high bone mass in mice. *Prrx1Cre; Zfp467*fl/fl but not *AdipoqCre; Zfp467*fl/fl mice exhibit high bone mass and greater osteogenic differentiation similar to the *Zfp467*-/- mice. qPCR results revealed that PTH suppressed *Zfp467* expression primarily via the cyclic AMP/PKA pathway. Not surprisingly, PKA activation inhibited the expression of *Zfp467* and gene silencing of *Pth1r* caused an increase in *Zfp467* mRNA transcription. Dual fluorescence reporter assays and confocal immunofluorescence demonstrated that genetic deletion of *Zfp467* resulted in higher nuclear translocation of NFκB1 that binds to the P2 promoter of the *Pth1r* and increased its transcription. As expected, *Zfp467*-/- cells had enhanced production of cyclic AMP and increased glycolysis in response to exogenous PTH. Additionally, the osteogenic response to PTH was also enhanced in *Zfp467*-/- COBs, and the pro-osteogenic effect of *Zfp467* deletion was blocked by gene silencing of *Pth1r* or a PKA inhibitor. In conclusion, our findings suggest that loss or PTH1R-mediated repression of *Zfp467* results in a pathway that increases *Pth1r* transcription via NFκB1 and thus cellular responsiveness to PTH/PTHrP, ultimately leading to enhanced bone formation.

## Editor's evaluation

The study provides evidence that the hormone PTH increases bone mass by, at least in part, regulating the factor Zfp467. In turn, Zfp67 controls expression of the receptor for PTH, thus creating a feedback loop that overall augments bone mass. The findings are novel and of great interest. The study is significant as it unveils a novel feedback loop involving PTH, a critical endocrine regulator of calcium, phosphate, and bone mass.

## Introduction

Intermittent administration of parathyroid hormone (PTH 1–34), teriparatide, is anabolic for bone and is a well-established treatment for osteoporosis (*Estell and Rosen, 2021*). Both PTH and PTHrP act through the PTH1R to drive bone formation, increase bone mass, and lower fracture risk (*Goltzman, 2018*). It has been established that one mechanism for PTH-driven osteogenesis is through induction of cyclic AMP and the PKA pathway, although PKC can also be activated by ligand binding to the PTH1R (*Wein and Kronenberg, 2018*). Downstream targets from PTH1R activation include IGF-1, FGF-2, Rankl, Sclerostin, Wnt signaling, salt-inducible kinases, and the BMPs (*Wein and Kronenberg, 2018*; *Nishimori et al., 2019*; *Yu et al., 2012*). *Pth1r* is expressed in chondrocytes, and the entire osteoblast lineage from early osteogenic progenitors to osteoblasts during which it is upregulated (*Goltzman, 2018*). Osteocytes and lining cells also express *Pth1r* (*Goltzman, 2018*; *Wein and Kronenberg, 2018*). Recent studies have noted that the PTH1R is also expressed in mature adipocytes and their immediate precursors, marrow adipocyte-like progenitors, or MALPs (*Zhong et al., 2020*).

Intermittent PTH treatment increases bone formation both by enhancing the number of osteoblasts and their function, resulting in higher bone mass (*Balani et al., 2017*). Several reports have demonstrated that PTH-induced lineage allocation of skeletal stromal cells into osteoblasts is at the expense of adipogenesis (*Fan et al., 2017*). This is consistent with human studies which confirmed that PTH treatment reduces bone marrow adiposity principally through a shift in lineage allocation (*Maridas et al., 2019*). The transcriptional mechanisms whereby PTH drives a progenitor cell toward an osteoblast are multiple, complex, and redundant. Previous study reported that genetic deletion of the PTH1R in Prrx1+ cells resulted in low bone mass and a marked increase in bone marrow adiposity (*Fan et al., 2017*). A previous study for the mechanisms that drive adipogenesis that identified *Zfp467* as a gene markedly upregulated in the absence of the PTH1R (*Fan et al., 2017*). Zinc finger proteins (ZFPs) are one of the largest classes of transcription factors in eukaryotic genomes and *Zfp521* and *Zfp423* have been identified as critical determinants of both adipogenesis and osteogenesis (*Ganss and Jheon, 2004*; *Kiviranta et al., 2013*). *Zfp467* has been reported to enhance both *Sost* and *Pparg* expression in marrow stromal cells, and is highly expressed in both adipocyte and osteoblast progenitors (*Quach et al., 2011*). Furthermore, an earlier study identified *Zfp467* as a transcriptional

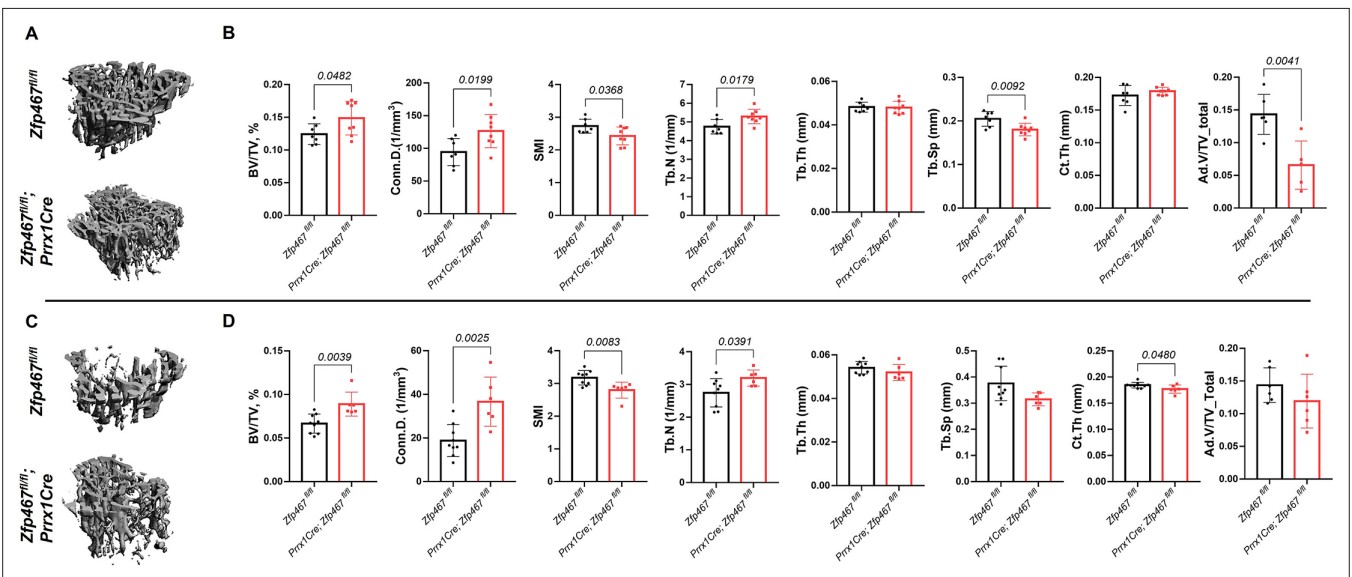

**Figure 1.** *Prrx1Cre Zfp467* mice have more trabecular bone mass and less adipose tissue in bone marrow, recapitulating the global Zfp467 null mice. (**A, B**) Male and (**C, D**) female 12-week-old *Prrx1Cre; Zfp467fl/fl* mice and control mice were measured using trabecular and cortical bone of tibiae. Marrow adipose tissue volume (Ad.V) was quantified by osmium tetroxide staining and micro-computed tomography (μCT). Data shown as mean ± SD by unpaired Student's t test, n=5–8 per group.

The online version of this article includes the following figure supplement(s) for figure 1:

**Figure supplement 1.** Generation of Zfp467 flox mice.

**Figure supplement 2.** *AdipoqCre Zfp467* have similar trabecular and cortical bone mass with controls.

regulator of lineage allocation among mesenchymal progenitors cells in the marrow (**You et al., 2012**). In accordance with that report, global genetic deletion of *Zfp467* resulted in high bone mass and a reduction in bone marrow adipose tissue (BMAT) and peripheral adipose depots (**Le et al., 2021**). Hence, the inverse relationship between PTH1R and *Zfp467* suggested an additional pathway by which PTH could impact lineage allocation. To determine the molecular mechanism of this interaction, *Zfp467*-/- osteoblast progenitors were used for studying and a potential transcriptional modulator of the PTH1R, regulated by ZFP467, was identified. This report details both in vitro and in vivo studies, and points to a novel pathway that impacts skeletal formation through the PTH1R.

## Results

Previous study showed that global deletion of *Zfp467* increased trabecular bone volume and cortical bone thickness, compared to wild-type mice at the age of 16 weeks (**Le et al., 2021**). And histomorphometry results showed higher structural and dynamic formation parameters in *Zfp467*-/- mice vs. *Zfp467*+/+ (**Le et al., 2021**). To assess whether the effect on bone mass seen in the *Zfp467* global knockout (KO) mice was cell-autonomous to mesenchymal cells, we generated mice with deletion of Zfp467 in the limb MSCs by crossing *Zfp467*fl/fl mice with the *Prrx1Cre* mice or *AdipoqCre* mice (**Figure 1—figure supplement 1**).

### *Similar to the* global Zfp467 null mice, *Prrx1Cre Zfp467* mice have increased trabecular bone mass

Body mass and body size were not significantly different between *Prrx1Cre; Zfp467*fl/fl and control littermates in both males and females (data not shown). Micro-computed tomography (μCT) analysis showed that *Prrx1Cre; Zfp467*fl/fl mice had higher trabecular bone volume fraction (Tb.BV/TV), greater connectivity density (Conn.D), and higher trabecular number (Tb.N) with a significant

**Table 1.** Quantification of structural and cellular parameters in the left tibiae of 12-week-old *Prrx1Cre;Zfp467*fl/fl and control *Zfp467*fl/fl mice by histomorphometry.

| | Male | | | Female | | |
|---|---|---|---|---|---|---|
| | *Zfp467*fl/fl | *Prrx1Cre; Zfp467*fl/fl | p-Value | *Zfp467*fl/fl | *Prrx1Cre; Zfp467*fl/fl | p-Value |
| BV/TV (%) | 9.1482±2.6523 | 12.570±2.9694 | 0.06 | 7.4561±1.9473 | 8.1315±3.8504 | 0.71 |
| Tb.Th (μm) | 35.961±3.7772 | 37.700±5.8760 | 0.63 | 35.384±2.5611 | 34.748±6.5862 | 0.76 |
| Tb.Sp (μm) | 396.80±168.61 | 274.83±62.817 | 0.13 | 484.18±119.44 | 501.35±288.61 | 0.90 |
| Tb.N (n/μm) | 2.5294±0.6037 | 3.3260±0.4942 | 0.03 | 2.0813±0.4175 | 2.3440±1.1650 | 0.61 |
| OS/BS (%) | 21.767±11.722 | 31.987±12.704 | 0.16 | 14.244±5.1159 | 8.8012±4.2810 | 0.07 |
| O.Th (μm) | 2.0084±0.4411 | 4.8227±3.4306 | 0.07 | 3.1650±0.4587 | 3.4638±1.2793 | 0.60 |
| Ob.S/BS (%) | 16.672±7.6973 | 25.817±8.5871 | 0.07 | 21.776±6.8886 | 18.210±8.0428 | 0.43 |
| N.Ob/B.Pm (n/mm) | 12.333±5.4003 | 19.082±5.6469 | 0.05 | 17.250±5.1870 | 14.051±4.7140 | 0.29 |
| Oc.S/BS (%) | 10.601±3.6508 | 12.975±4.2334 | 0.31 | 18.439±4.7740 | 19.192±6.7888 | 0.83 |
| N.Oc/B.Pm (n/mm) | 5.3885±1.9701 | 6.3505±2.1171 | 0.42 | 8.7722±2.1935 | 9.6039±3.4546 | 0.63 |
| MS/BS (%) | 45.655±3.5366 | 43.990±4.3419 | 0.47 | 45.333±5.4009 | 44.027±1.8733 | 0.68 |
| MAR (μm/day) | 1.2265±0.2238 | 1.1889±0.1266 | 0.71 | 1.8148±0.1294 | 1.6824±0.4816 | 0.63 |
| BFR/BS (μm³/μm³/day) | 0.5634±0.1373 | 0.5220±0.0648 | 0.49 | 0.8207±0.0933 | 0.7457±0.2292 | 0.59 |
| BFR/BV (%/day) | 3.2079±1.2339 | 2.6756±0.5373 | 0.32 | 4.6232±0.4123 | 4.5078±1.7235 | 0.85 |

Data are means ± SD (n=6–7). BV/TV, bone volume/total volume; Tb.Th, trabecular thickness; Tb.Sp, trabecular separation; Tb.N, trabecular number; OS/BS, osteoid surface/bone surface; O.Th, osteoid thickness; Ob.S/BS, osteoblast surface/bone surface; N.Ob/B.Pm, osteoblast number/bone perimeter; Oc.S/BS, osteoclast surface/bone surface; N.Oc/BS, osteoclast number/ bone surface; N.Oc/B.Pm, osteoclast number/bone perimeter; MS/BS, mineralizing surface/ bone surface; MAR, mineral apposition ratio; BFR/BS, bone formation ratio/ one surface; BFR/BV, bone formation rate/bone surface.

decrease in structural model index (SMI) and trabecular separation (Tb.Sp) in both males and females (*Figure 1*), indicating an increase in trabecular bone mass. Cortical thickness was increased, although not significantly in males and marginally decreased in females at this age. In contrast, tibial adipose tissue volume fraction in the marrow was significantly decreased in males (*Figure 1B*), and showed a similar non-significant trend in females (*Figure 1D*). Although the time point of examination differed (12 weeks here instead of 16 weeks in the global KO; *Le et al., 2021*), these results show that, overall, *Prrx1Cre; Zfp467*fl/fl mice showed a similar bone phenotype to the global *Zfp467* KO mice. It is therefore conceivable that progenitor fate was indeed altered in the global *Zfp467* KO mice, resulting in significant beneficial skeletal changes.

To assess the cellular activities that contributed to increased bone mass in male *Prrx1Cre;Zfp467*fl/fl mice, static and dynamic histomorphometry was performed. Consistent with μCT, histomorphometric analysis showed that although the changes in BV/TV were not significant, the overall trend was clearly toward an increase, and Tb.N (p=0.0271) and number of osteoblasts/total area (N.Ob/T.Ar, p=0.0105) were markedly increased in *Prrx1Cre;Zfp467*fl/fl male mice compared to *Zfp467*fl/fl control (*Table 1*), confirming an increase in osteogenesis and bone mass. However, in contrast to global *Zfp467* KO mice, the increased trabecular bone was observed in male *Prrx1Cre;Zfp467*fl/fl mice only, whereas no significant changes in the bone phenotype were found in female mice between groups.

## *AdipoqCre Zfp467* mice have similar cortical and trabecular bone mass to controls

In order to determine the whole-body phenotype of the conditional knockout (cKO) of *Zfp467* mice. *AdipoqCre Zfp467*fl/fl; mice have similar body weight, fat mass, lean mass, and femoral areal BMD at 12 weeks compared to control mice in both males and females (data not shown). μCT was performed and analyzed in the metaphysis and cortical bone at the tibial mid-diaphysis in 12-week-old *AdipoqCre Zfp467*fl/fl; mice and control mice. Not surprisingly, no significant difference was found regarding Tb.BV/TV, Conn.D, Tb.N, trabecular thickness (Tb.Th), Tb.Sp, and SMI (*Figure 1—figure supplement 2A,B*) between controls and *AdipoqCre* mice. In addition, no differences in cortical bone measurements including total area (Tt. Ar), cortical area/total area (Ct.Ar/Tt.Ar), cortical thickness (Ct.Th), marrow area (Ma.Ar), cortical porosity (Ct. Porosity), and cortical tissue mineral density (Ct.TMD) was observed between groups (*Figure 1—figure supplement 2A,B*).

## PTH suppressed the expression levels of *Zfp467* via the PKA pathway

To understand the mechanisms that underly the impact of mesenchymal deletion of Zfp467, the signaling pathway of PTH relative to Zfp467 in osteoblasts was examined by qPCR. Consistent with a previous study which showed that short-term PTH treatment suppressed the expression of *Zfp467* (*Quach et al., 2011*), treating cells with 100 nM PTH for 10 min could significantly suppress *Zfp467* expression in both COBs (*Figure 2A*) and bone marrow stromal cells (BMSCs) (*Figure 2B*). When pretreating COBs and BMSCs with PKA and PKC inhibitors (10 μM H89 and 5 μM Go6983, Selleck Chemicals, Houston, TX), respectively, for 2 hr prior to 10 min of 100 nM PTH treatment, significant rescue of *Zfp467* suppression was seen in H89 group, but Go6983 had no effect (*Figure 2C and D*). Forskolin, a selective PKA pathway activator, was also found to significantly inhibit the expression level of *Zfp467* in COBs (*Figure 2E*) and BMSCs (*Figure 2F*). Moreover, consistent with previous study that *Prrx1Cre; Pth1r*fl/fl mice showed higher expression level of *Zfp467* in bone marrow, silenced *Pth1r* with siRNA in both COBs and BMSC, and found *Pth1r* knockdown significantly upregulated the expression level of *Zfp467* (*Figure 2G and H*). These data suggested that PTH1R activation could downregulate *Zfp467* expression via the PKA pathway, and that ZFP467 could be one of the important downstream targets of PTH signaling.

## *Zfp467*-/- cells have greater *Pth1r* transcriptional levels driven by both the P1 and P2 promoter

Our previous study showed that the global absence of *Zfp467* resulted in a significant increase in trabecular bone volume, a marked reduction in peripheral and marrow adipose tissue, and an ~40% increase in *Pth1r* gene expression in bone from the *Zfp467*-/- mice compared to littermate controls (*Le et al., 2021*). These data suggested the possibility of a positive feedback loop whereby the suppression of *Zfp467* mediated by PTH leads to an increase of PTH1R. Consistent with this tenet, higher

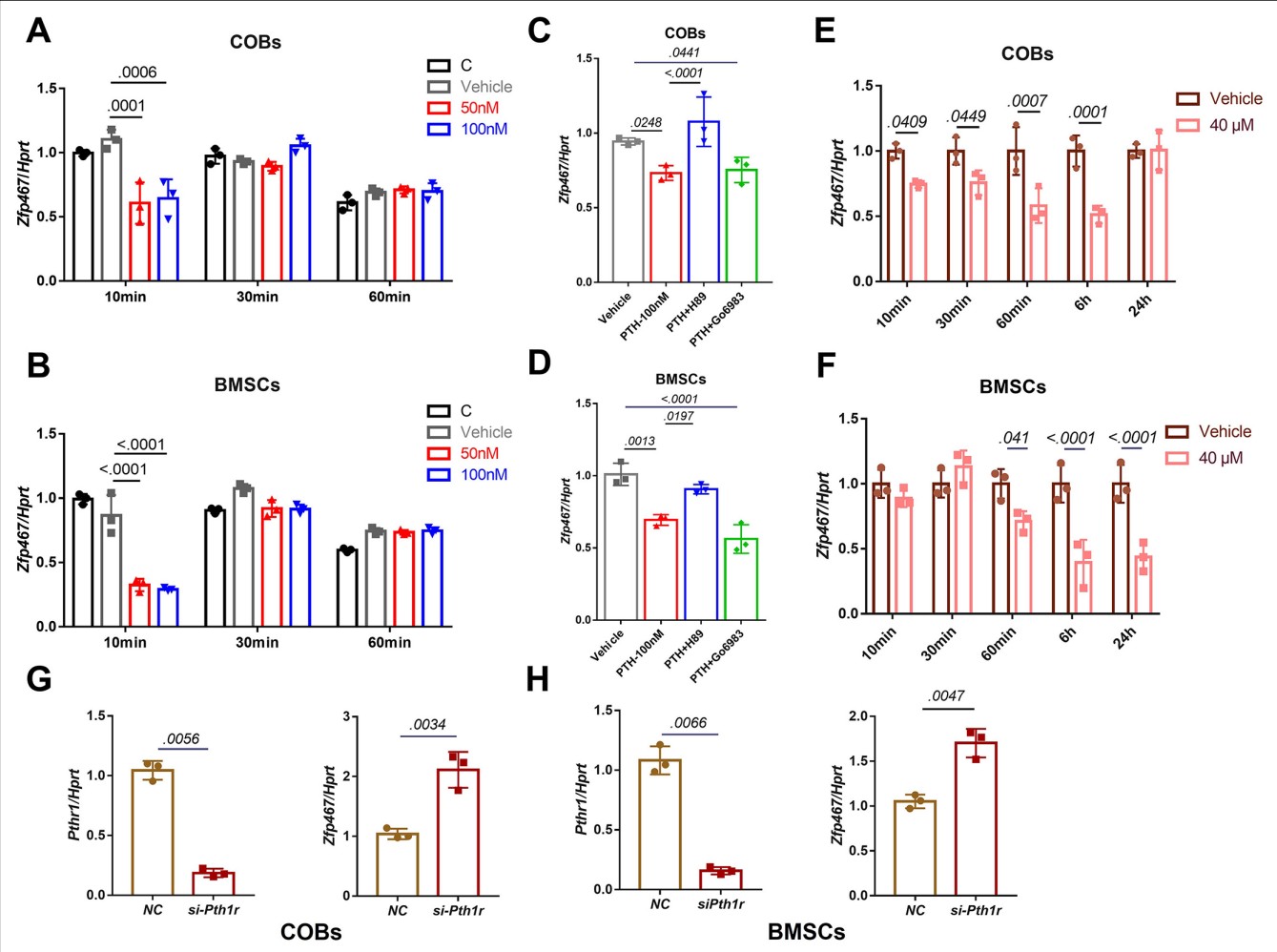

**Figure 2.** Parathyroid hormone (PTH) suppressed the expression levels of *Zfp467* via the PKA pathway. (**A**) PTH treatments significantly suppressed *Zfp467* expression within 10 min of treatment in *Zfp467*[+/+] calvarial osteoblasts (COBs). Data shown as mean ± SD by one-way ANOVA, n=3 independent experiments for each group. (**B**) PTH treatments significantly suppressed *Zfp467* expression within 10 min of treatment in *Zfp467*[+/+] bone marrow stromal cells (BMSCs). Data shown as mean ± SD by one-way ANOVA, n=3 independent experiments for each group. (**C**) qPCR results· of *Zfp467*[+/+] in COBs with 2 hr PKA or PKC inhibitor treatment prior to 10 min of 100 nM PTH exposure, PKA but not PKC inhibitor was able to rescue the suppression of *Zfp467* induced by PTH. Data shown as mean ± SD by one-way ANOVA, n=3 independent experiments for each group. (**D**) qPCR results of *Zfp467*[+/+] in BMSCs with 2 hr PKA or PKC inhibitor treatment prior to 10 min of 100 nM PTH exposure. PKA but not PKC inhibitor was able to rescue the suppression of *Zfp467* induced by PTH. Data shown as mean ± SD by one-way ANOVA, n=3 independent experiments for each group. Forskolin significantly suppressed *Zfp467* expression within 1 hr of treatment in *Zfp467*[+/+] COBs. Data shown as mean ± SD by unpaired Student's t test, n=3 independent experiments for each group. (**F**) Forskolin significantly suppressed *Zfp467* expression after 6 hr of treatment in *Zfp467*[+/+] BMSCs. Data shown as mean ± SD by unpaired Student's t test, n=3 independent experiments for each group. (**G**) *Pth1r*-siRNA treatment in *Zfp467*[+/+] COBs led to an increase of *Zfp467* expression in *Zfp467*[+/+] COBs. Data shown as mean ± SD by unpaired Student's t test, n=3 independent experiments for each group. (**H**) *Pth1r*-siRNA treatment in *Zfp467*[+/+] COBs led to an increase of *Zfp467* expression in *Zfp467*[+/+] BMSCs. Data shown as mean ± SD by unpaired Student's t test, n=3 independent experiments for each group. NC, negative control.

gene and protein expression level of PTH1R was found in *Zfp467*[-/-] COBs and BMSCs (*Figure 3A and B*). Three transcripts of *Pth1r* (NM_011199.2, NM_001083935.1, and NM_001083936.1) with different transcription starting sites (TSS) were reported based on NCBI database and UCSC Genome Browser on Mouse (*Figure 3C*). The three transcripts shared the same coding sequence and the only difference was located at the 5' untranslated region. Based on the different 5' untranslated regions, related primers were designed; by qPCR both the *Pth1r-T1* and *Pth1r-T2* transcripts were upregulated in *Zfp467*[-/-] cells (*Figure 3D*). However, the expression level of *Pth1r-T2* was much higher than *Pth1r-T1*. *Pth1r-T1* and *Pth1r-T2* transcripts were driven by P1 and P2 promoters, respectively. As P1 is much longer than P2 and hadn't been investigated before, a P1 promoter-driven dual-fluorescence reporter

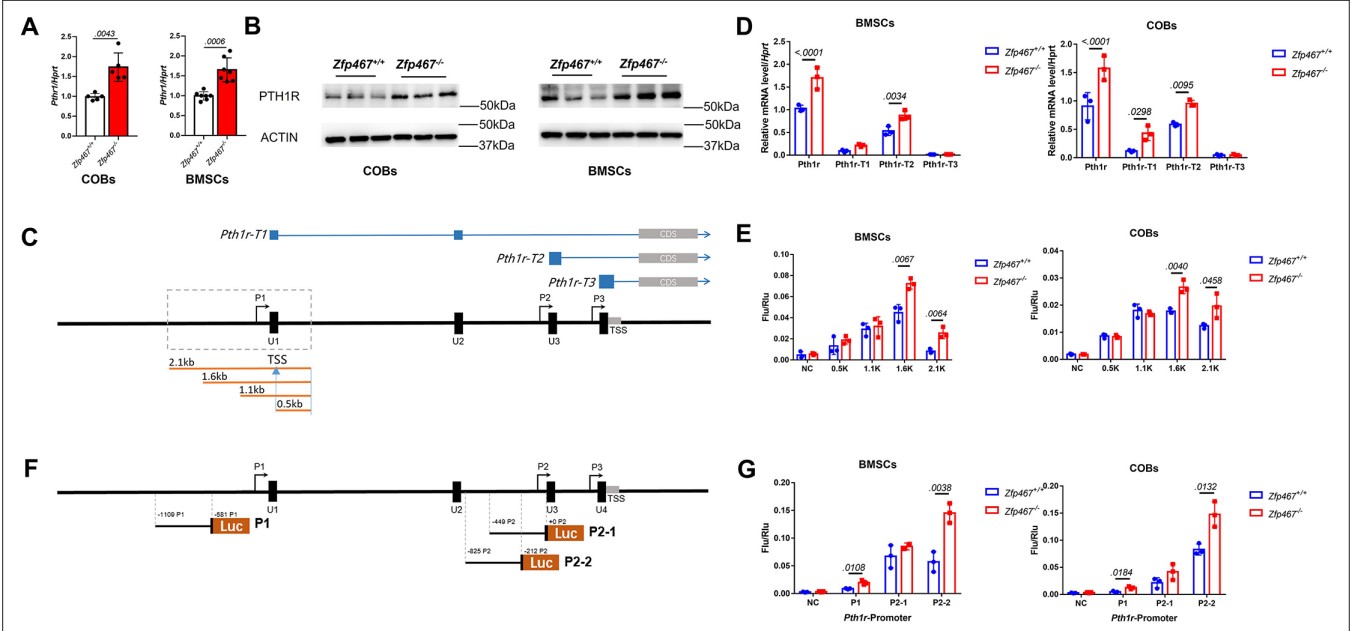

**Figure 3.** *Zfp467-/-* cells have greater *Pth1r* transcriptional levels driven by both the P1 and P2 promoter. (**A**) qPCR results of baseline calvarial osteoblasts (COBs) and bone marrow stromal cells (BMSCs). Higher expression level of *Pth1r* was found in both *Zfp467-/-* COBs and BMSCs. Data shown as mean ± SD by unpaired Student's t test, n=5–7 independent experiments for each group. (**B**) Western blot analysis of baseline COBs and BMSCs. Higher expression level of PTH1R was found in both *Zfp467-/-* COBs and BMSCs. (**C**) A schematic of three different *Pth1r* transcripts and P1 promoter of *Pth1r*. Four different length P1 promoter constructs were designed and inserted into dual-fluorescence reporter vector. (**D**) qPCR results of three *Pth1r* transcripts and total *Pth1r*. Total *Pth1r* and *Pth1r-T1, T2* but not *Pth1r-T3* were upregulated in both *Zfp467-/-* COBs and BMSCs. Data shown as mean ± SD by unpaired Student's t test, n=3 independent experiments for each group. (**E**) Dual-fluorescence assay using indicated four P1 reporter constructs. The 1.6 and 2.1 kb constructs-driven reporter is higher activated in *Zfp467-/-* cells compared to *Zfp467+/+* cells. Data shown as mean ± SD by unpaired Student's t test, n=3 independent experiments for each group. (**F**) A schematic of P1 and P2 promoter constructs of *Pth1r*. (**G**) Dual-fluorescence assay using indicated P1 and P2 reporter constructs. Both P1 and P2-2 were found significantly higher activated in *Zfp467-/-* cells. Data shown as mean ± SD by unpaired Student's t test, n=3 independent experiments for each group. NC, negative control, TSS, transcription starting site.

The online version of this article includes the following source data for figure 3:

**Source data 1.** Western blot for *Figure 3B*.

**Source data 2.** Western blot for *Figure 3B* PTH1R in calvarial osteoblasts (COBs).

**Source data 3.** Western blot for *Figure 3B* ACTIN in calvarial osteoblasts (COBs).

**Source data 4.** Western blot for *Figure 3B* PTH1R in bone marrow stromal cells (BMSCs).

**Source data 5.** Western blot for *Figure 3B* ACTIN in bone marrow stromal cells (BMSCs).

with four different length P1 promoters were designed to assess any change in the promoter activity of P1 in both COBs and BMSCs in the absence of 467 (*Figure 3C*). The 1.6 and 2.1 kb promoter-driven reporters are higher in activated *Zfp46 -/-* cells compared to *Zfp467+/+* cells (*Figure 3E*), which indicated the binding site of *Pth1r* P1 promoter in *Zfp467 -/-* cells is between 0.6 and 1.1 kb ahead of P1 TSS.

Combined with the previously reported two potential transcription factor binding sites of P2 (*Tohmonda et al., 2013*), three P1- and P2-driven dual-fluorescence reporters were constructed (*Figure 3F*). Both P1 and P2 were activated in *Zfp467-/-* cells, but P2 showed much higher activity in BMSCs and COBs. P2-2 was also much more active in *Zfp467-/-* cells than *Zfp467+/+* cells. P2-2 was therefore chosen for transcription factor prediction via JASPAR, Animal TFDB and PROMO databases; six transcription factors predicted by all three databases were noted, including CREB, EBF1, MYOD, cFOS, NFκB1, and GATA1.

## *Zfp467-/-* cells have higher NFκB1 and GATA1 nuclear translocation

Using qPCR, no differences in the transcriptional levels of these transcription factors was observed (*Figure 4—figure supplement 1*). Therefore, those potential transcription factors that could bind to

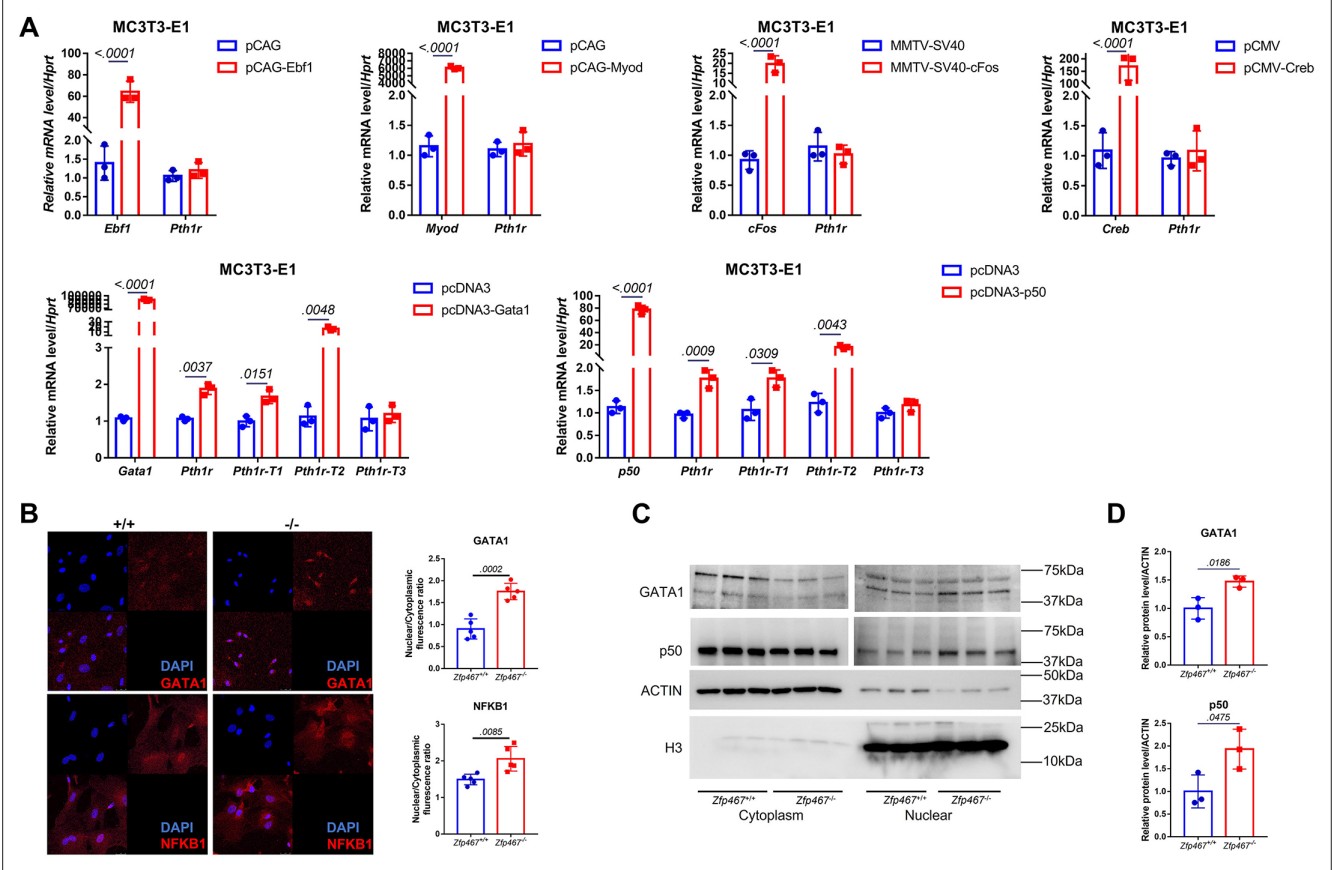

**Figure 4.** *Zfp467⁻/⁻* cells have higher NFκB1 and GATA1 nuclear translocation. (**A**) qPCR results of overexpression of *Ebf1*, *Myod*, *Myog*, *Gata1*, and *NFκB1* in MC3T3-E1 cell line. GATA1 and NFκB1 overexpression could significantly upregulate the expression level of *Pth1r*. Data shown as mean ± SD by unpaired Student's t test, n=3 independent experiments for each group. (**B**) Representative confocal images of GATA1 and NFκB1 immunofluorescence in *Zfp467⁺/⁺* and *Zfp467⁻/⁻* bone marrow stromal cells (BMSCs) and related quantification. (**C**) Nuclear protein level of GATA1 and NFκB1 in *Zfp467⁺/⁺* and *Zfp467⁻/⁻* BMSCs. (**D**) Quantification analysis for nuclear protein level of GATA1 and NFκB1 in *Zfp467⁺/⁺* and *Zfp467⁻/⁻* BMSCs. Data shown as mean ± SD, n=3 independent experiments for each group. Data shown as mean ± SD by unpaired Student's t test, n=3 independent experiments for each group.

The online version of this article includes the following source data and figure supplement(s) for figure 4:

**Source data 1.** Western blot for *Figure 3C*.

**Source data 2.** Western blot for *Figure 3C* ACTIN.

**Source data 3.** Western blot for *Figure 3C* H3.

**Source data 4.** Western blot for *Figure 3C* p50 in cytoplasm.

**Source data 5.** Western blot for *Figure 3C* GATA1 in cytoplasm.

**Source data 6.** Western blot for *Figure 3C* GATA1 in nuclear.

**Source data 7.** Western blot for *Figure 3C* p50 in nuclear.

**Figure supplement 1.** qPCR results of expression of *Ebf1*, *Myod*, *Fos*, *Gata1*, *Nfkb1*, and *Creb* in *Zfp467⁺/⁺* and *Zfp467⁻/⁻* calvarial osteoblasts (COBs) and bone marrow stromal cells (BMSCs).

cryptic sites in the *Pth1r* P2-2 promoter in a pre-osteoblast cell line MC3T3-E1 were overexpressed subsequently. Only GATA1 and NFκB1 overexpression could significantly upregulate the expression level of *Pth1r*, especially *Pth1r-T1* and *-T2* (**Figure 4A**). Nuclear translocation level of NFκB1 and GATA1 were further detected using immunofluorescence, nuclear protein isolation, and western blot. NFκB1 and GATA1 were almost evenly distributed in the cytoplasm and nucleus of *Zfp467⁺/⁺* cells but underwent partial translocation to the nucleus in *Zfp467⁻/⁻* cells (**Figure 4B**). *Zfp467⁻/⁻* cells also showed much higher nuclear protein level of both NFκB1 and GATA1 (**Figure 4C and D**).

## An NFκB1-RelB heterodimer may drive greater *Pth1r* transcription in *Zfp467*[-/-] cells

In order to confirm whether NFκB1 or GATA1 could activate the specific P1 or P2-2 Pth1r promoter, P1 and P2-2 dual-luciferase reporter and *Gata1*, *Nfkb1* overexpression plasmids were co-transfected in MC3T3-E1 cells. Only the *Nkfb1* overexpression group was able to significantly activate the P2-2 promoter (**Figure 5A**). Chromatin immunoprecipitation (ChIP) results showed that the DNA was properly sheared and IP was successfully conducted (**Figure 5B**). ChIP-qPCR results showed that the first two parts of P2 were properly enriched in our IP product (>0.5%) (**Figure 5—figure supplement 1A**), and the first part of P2 was approximately 20-fold more highly enriched in our NFκB1 IP product than IgG (**Figure 5C**, **Figure 5—figure supplement 1B**); this indicated that NFκB1 binds to the P2 promoter, especially at the first 200 bp site. Subsequently, COBs and BMSCs were treated with *Nfkb1* siRNA and showed that *Nfkb1* knockdown could significantly inhibit the expression of *Pth1r* in both *Zfp467*[+/+] and *Zfp467*[-/-] COBs and BMSCs. Importantly, *Nfkb1* knockdown in *Zfp467*[-/-] cells reverts the levels of *Pth1r* to the levels seen in *Zfp467*[+/+] cells (**Figure 5D–G**).

In order to determine whether NFκB1 could bind to the Pth1r P2-2 promoter directly, DNA pulldown assay was performed using biotin-labeled *Pth1r P2* promoter as a probe. As shown in **Figure 6A**, the biotin-Pth1rP2 group showed a specific band in both MC3T3-E1 nuclear extracts and purified NFκB1 protein, suggesting a direct physical interaction between NFκB1 and Pth1r P2 promoter. However, noticed that NFκB1 does not have a transcriptional activation domain, NFκB1 must heterodimerize with other transcription factors in order to increase gene transcription. Using String database and checking published studies, eight candidates that might heterodimerize with NFκB1 to regulate gene transcription were obtained: NFYC, NPAS1, Rel, AKAP8, RelA, RelB, ANKRD42, and HDAC1. Using siRNA to knock down all these potential NFκB1 partners (**Figure 6B**), the upregulated *Pth1r* induced by *Nfkb1* overexpression could only be dampened by *Npas1* and *Relb* siRNA (**Figure 6C**). Further co-immunoprecipitation (co-IP) results confirmed that NFκB1 could heterodimerize with RelB only (**Figure 6D**), which suggested that NFκB1-Relb heterodimers may drive greater *Pth1r* transcription in *Zfp467*[-/-] cells.

Given that post-translational regulation is essential for NF-κB activity, whether more proximal events in the NF-κB signaling cascade are impacted in Zfp467-deficient cells were also determined. p50, p105, and p105 phosphorylation was measured in *Zfp467* siRNA-treated MC3T3-E1 cells; no significant difference between groups was observed (**Figure 6E**). Therefore, ZFP467 may have no effect on p105 proteasomal processing and p50 production.

In the non-canonical NFκB pathway, NFκB-inducing kinase (NIK) is constitutively degraded by the proteasome and this process is TRAF3 and TRAF2-dependent. And when non-canonical NFκB signaling is activated, TRAF2 and TRAF3 are recruited to the receptors, TRAF3 will be degraded and NIK stabilization is achieved. Accumulated NIK then phosphorylates IKKα and helps IKKα to phosphorylate p100. Phosphorylated p100 will be degraded followed by ubiquitination and relieve RelB, then RelB can heterodimerize with p50 and be transported to the nucleus. Furthermore, higher protein levels of NIK was found in *Zfp467* knockdown cells, but there was no difference in its transcriptional level (**Figure 6F**). Moreover, in *Zfp467* knockdown cells, higher phosphorylation level of both IKKα and p100 was observed (**Figure 6G**), which suggested the non-canonical NFκB pathway was activated. Although slightly higher transcriptional level of *Ikka* was found, there were no differences in its protein level.

In order to investigate whether there was a stimulus between Zfp467 and non-canonical NFκB pathway, the expression level of a few classic non-canonical NFκB pathway stimuli and related receptors in *Zfp467* siRNA-treated MC3T3-E1 cells were detected; we found *Rankl, Ltb,* and *Baff* was much higher in *Zfp467* siRNA-treated MC3T3-E1 cells than control cells (**Figure 6—figure supplement 1**).

## *Zfp467*[-/-] cells have increased PTH signaling and higher extracellular acidification rates

As the PKA pathway is considered to be one of the major downstream pathways of the PTH signaling network, it would be important to know whether *Zfp467* [-/-] cells have higher PKA activation levels due to the higher expression of PTH1R. PTH increased cAMP within 10 min, measured by ELISA in *Zfp467*[+/+] COBs, and *Zfp467*[-/-] cells had a higher levels of cAMP expression than *Zfp467*[+/+] cells (p=0.0007 for vehicle, p<0.0001 for 50 nM, p=0.0008 for 100 nM). In addition, *Zfp467*[-/-] BMSCs also had significantly

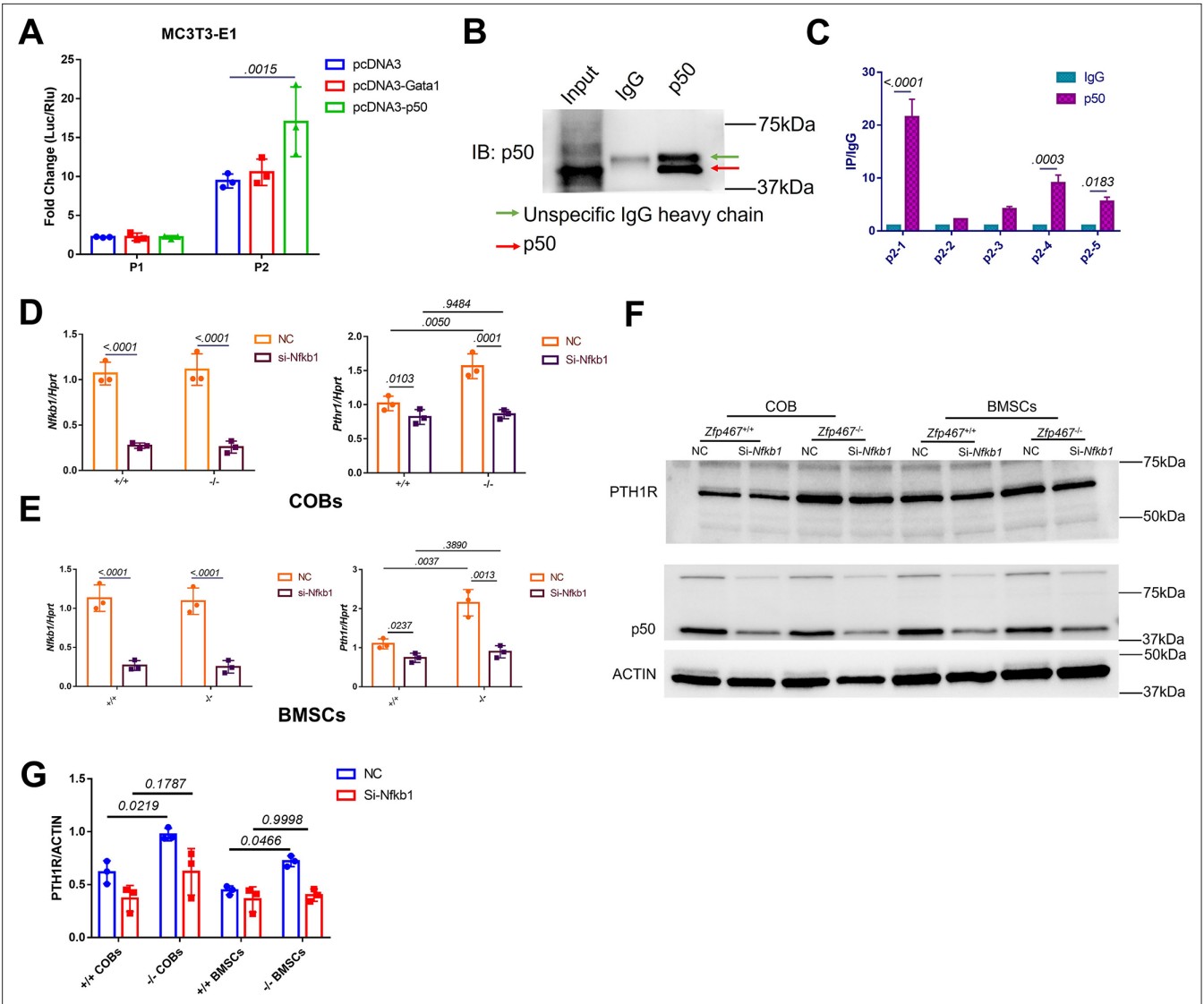

**Figure 5.** NF$\kappa$B1 was found to transactivate *Pth1r* expression in *Zfp467*$^{-/-}$ cells. (**A**) Reporter assays using the indicated P1 or P2 reporter construct and an expression vector bearing *Gata1*, *Nfkb1*, or a control empty vector. Data shown as mean ± SD by one-way ANOVA, n=3 independent experiments for each group. Data shown as mean ± SD by one-way ANOVA, n=3 independent experiments for each group. (**B**) Immunoblot assay using a control rabbit IgG antibody (IgG) or the anti-NF$\kappa$B1 antibody during chromatin immunoprecipitation assay. (**C**) DNA enrichment of Pth1r P2 promoter, ratio between NF$\kappa$B1 and IgG IP products, first part and last two parts of P2 were significantly enriched by NF$\kappa$B1 antibody. Data shown as mean ± SD by unpaired Student's t test, n=3 independent experiments for each group. (**D, E**) qPCR results of the expression levels of *Nfkb1* and *Pth1r* in *Nfkb1* siRNA-treated *Zfp46*$^{+/+}$ and *Zfp467*$^{-/-}$ calvarial osteoblasts (COBs) and bone marrow stromal cells (BMSCs). Data shown as mean ± SD by two-way ANOVA, n=3 independent experiments for each group. (**F**) Western blot analysis of *Nfkb1* and *Pth1r* in *Nfkb1* siRNA-treated *Zfp467*$^{+/+}$ and *Zfp467*$^{-/-}$ COBs and BMSCs. (**G**) Quantification for PTH1R protein level. Data shown as mean ± SD by two-way ANOVA, n=3 independent experiments for each group. NC, negative control.

The online version of this article includes the following source data and figure supplement(s) for figure 5:

**Source data 1.** Western blot for *Figure 5B and F*.

**Source data 2.** Western blot for *Figure 5B* p50 with IP samples.

**Source data 3.** Western blot for *Figure 5C* PTH1R.

**Source data 4.** Western blot for *Figure 5C* p50.

**Source data 5.** Western blot for *Figure 5C* ACTIN.

**Figure supplement 1.** p2 promoter of Pth1r was properly enriched in p50-ChIP-qPCR assay using MC3T3-E1 nuclear extracts.

**Figure supplement 1—source data 1.** Nucleic acid blot for *Figure 5—figure supplement 1B*.

**Figure supplement 1—source data 2.** Nucleic acid blot for *Figure 5—figure supplement 1B*, PCR blot for Pth1r amplification product.

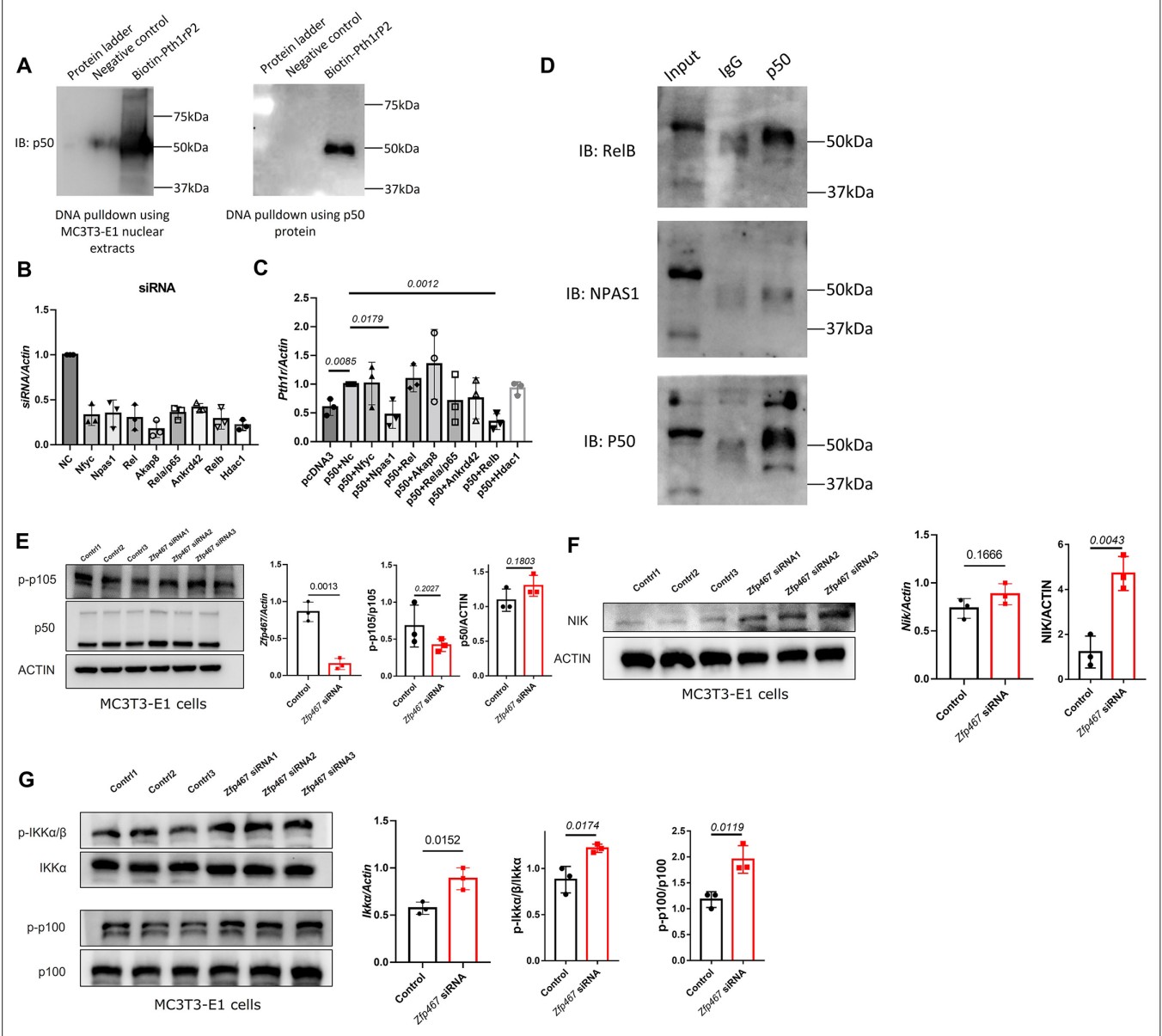

**Figure 6.** NF κ B1 heterodimerize with RelB to transactivate the expression of *Pth1r*. (**A**) DNA pulldown assay with biotin-labeled Pth1r P2. MC3T3-E1 nuclear extracts or NF κ B1 recombinant protein was probed with biotin-Pth1rP2 and then subjected to immunoblotting using NF κ B1 antibody. (**B**) qPCR results of the expression levels of *Nfyc*, *Npas1*, *Rel*, *Akap8*, *Rela*, *Ankrd42*, *Relb,* and *Hdac1* in related siRNA-treated MC3T3-E1 cells. Data shown as mean ± SD by one-way ANOVA, n=3 independent experiments for each group. NC, negative control. (**C**) qPCR results of the expression levels of *Pth1r* in *Nfkb1* overexpression plasmid and *Nfyc*, *Npas1*, *Rel*, *Akap8*, *Rela*, *Ankrd42*, *Relb*, or *Hdac1* siRNA co-transfected MC3T3-E1 cells. NC, negative control. (**D**) IP results using NF κ B1 antibody in MC3T3-E1 protein extracts, IgG was used as a negative control. (**E**) Protein level of p-p105 and p50 in *Zfp467* knockdown MC3T3-E1 cells. Data shown as mean ± SD by unpaired Student's t test, n=3 independent experiments for each group. (**F**) Protein and mRNA level of NF κ B-inducing kinase (NIK) in control and *Zfp467* siRNA-treated MC3T3-E1 cells. Data shown as mean ± SD by unpaired Student's t test, n=3 independent experiments for each group. (**G**) Protein level of p-IKKa and p-p100 and mRNA level of *Ikka* in control and *Zfp467* siRNA-treated MC3T3-E1 cells. Data shown as mean ± SD by unpaired Student's t test, n=3 independent experiments for each group.

The online version of this article includes the following source data and figure supplement(s) for figure 6:

**Source data 1.** Western blot for *Figure 6A, D, E, F and G*.

**Source data 2.** Western blot for *Figure 6A* p50 in DNA pulldown experiment using nuclear extract.

**Source data 3.** Western blot for *Figure 6A* p50 in DNA pulldown experiment using p50 purified protein.

**Source data 4.** Western blot for *Figure 6D* RelB.

**Source data 5.** Western blot for *Figure 6D* NPAS1.

*Figure 6 continued on next page*

*Figure 6 continued*

**Source data 6.** Western blot for *Figure 6D* p50.

**Source data 7.** Western blot for *Figure 6E* p-p105.

**Source data 8.** Western blot for *Figure 6E* p50/p105.

**Source data 9.** Western blot for *Figure 6E* ACTIN.

**Source data 10.** Western blot for *Figure 6F* NF κ B-inducing kinase (NIK).

**Source data 11.** Western blot for *Figure 6F* ACTIN.

**Source data 12.** Western blot for *Figure 6F* p-IKKα/β.

**Source data 13.** Western blot for *Figure 6F* IKKα.

**Source data 14.** Western blot for *Figure 6G* p-p100.

**Source data 15.** Western blot for *Figure 6G* p100.

**Figure supplement 1.** mRNA level of NF κ B non-canonical pathway stimulus and related receptors in control and *Zfp467* siRNA-treated MC3T3-E1 cells.

---

higher levels of intracellular cAMP after 10–60 min exposure of PTH ($p<0.0001$ for 10 min-100 nM, $p=0.0341$ for 30 min-100 nM, $p<0.0001$ for 60 min-100 nM) (*Figure 7A and B*). Importantly, *Zfp467*$^{-/-}$ COBs and BMSCs showed a greater magnitude of increase of cAMP after PTH treatment ($p=0.0053$ for interaction in COBs, $p=0.0010$ for interaction in BMSCs), which resulted from higher PTH1R in *Zfp467*$^{-/-}$ cells. Additionally, as CREB was one of the major downstream targets of PKA, higher p-CREB in *Zfp467*$^{-/-}$ COBs and BMSCs was observed whereas the total protein level of CREB showed no difference (*Figure 7C and D*).

In a previous study, PTH was shown to enhance aerobic glycolysis, which is a major source of ATP for osteoblast differentiation (*Esen et al., 2015*). The oxygen consumption and extracellular acidification in both COBs and BMSCs was measured pre-osteogenic differentiation and 3 days after osteogenic differentiation. Cellular respiration measurements showed that *Zfp467*$^{-/-}$ pre-differentiated COBs had significantly increased extracellular acidification rates (ECAR), but no difference was found for pre-differentiated BMSCs in respect to either ECAR or oxygen consumption rate (OCR) (*Figure 7E and F*). However, both *Zfp467*$^{-/-}$ COBs and BMSCs cells had significantly higher ECAR levels after 3 days of osteogenic differentiation, although no difference was found regarding OCR between genotypes (*Figure 7G and H*). These data suggest that *Zfp467*$^{-/-}$ COBs or BMSCs may have higher glycolysis as a result of higher PTH1R expression.

## *Zfp467*$^{-/-}$ cells showed increased sensitivity to PTH and enhanced pro-osteogenic as well as anti-adipogenic effects

To determine the sensitivity of *Zfp467*$^{-/-}$ cells to PTH, COBs were treated with osteogenic differentiation media and PTH for 7 or 14 days simultaneously. A dose response to PTH led to a significant increase in alkaline phosphatase (ALP) staining in *Zfp467*$^{-/-}$ COBs, and in *Zfp467*$^{+/+}$ cells. Furthermore, 100 nM PTH in *Zfp467*$^{-/-}$ cells produced remarkably higher positive-stained cells than vehicle group ($p=0.0001$) while there was no statistical significance seen among *Zfp467*$^{+/+}$ groups ($p=0.6536$) (*Figure 8A and B*). Importantly, *Zfp467*$^{-/-}$ COBs showed a more significant response to PTH regarding ALP and alizarin red staining ($p=0.0201$ for ALP, $p=0.0210$ for ARS) (*Figure 8B*). ARS showed a parallel trend as ALP staining; an increase in PTH dose resulted in an increase in mineralization for *Zfp467*$^{-/-}$ COBs only (*Figure 8A*).

In osteogenic media, lipid stored in osteoblasts was observed. Significantly less lipid droplets in *Zfp467*$^{-/-}$ osteoblasts were noted compared to untreated *Zfp467*$^{+/+}$ samples (*Figure 8C*). Additionally, a reduction in lipid droplet formation was observed with PTH exposure at 50 and 100 nM in both *Zfp467*$^{+/+}$ and *Zfp467*$^{-/-}$ cells, and *Zfp467*$^{-/-}$ cells showed much less lipid droplet formation compared to +/+ group (*Figure 8C and D*). However, the magnitude of decrease after PTH treatment is almost identical in *Zfp467*$^{+/+}$ and *Zfp467*$^{-/-}$ cells (*Figure 8D*). These results were then confirmed by qRT-PCR after 7 days' osteogenic differentiation which indicated higher expression of osteogenic differentiation-related genes such as *Alp, Sp7, Rankl,* and *Igf1* in *Zfp467*$^{-/-}$ compared to *Zfp467*$^{+/+}$ COBs with PTH exposure (*Figure 8E and F*). It is noteworthy that there is a statistically significant interaction between genotype and PTH treatment, suggesting an increased sensitivity to PTH in *Zfp467*$^{-/-}$ cells.

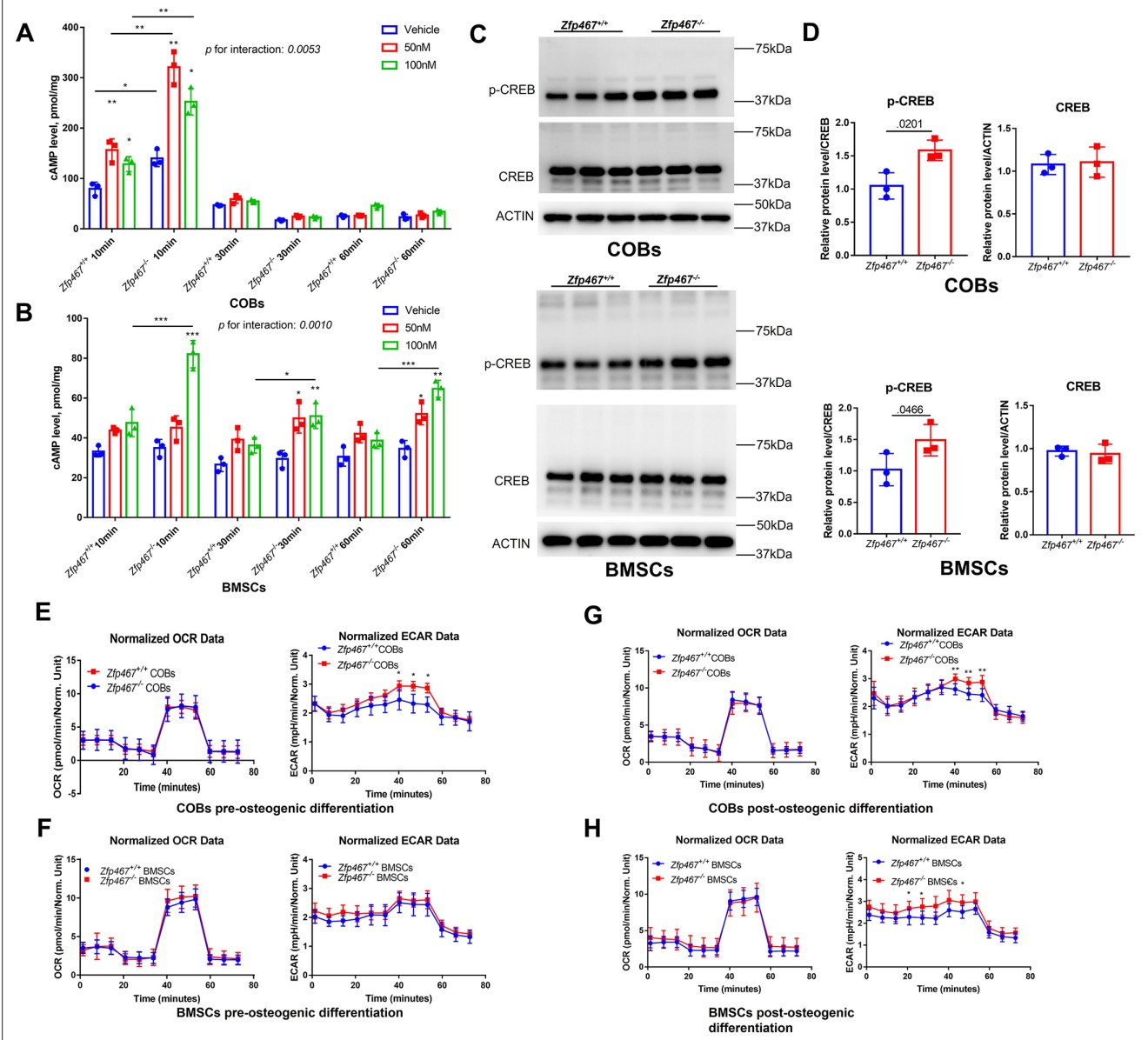

**Figure 7.** *Zfp467⁻/⁻* cells have increased parathyroid hormone (PTH) signaling and higher extracellular acidification rates. (**A**) Cyclin adenosine monophosphate (cAMP) ELISA of undifferentiated calvarial osteoblasts (COBs), from 10 to 60 min with 100 nM PTH treatment, PTH increased cAMP expression within 10 min of treatment, and *Zfp467⁻/⁻* cells have higher level of cAMP than *Zfp467⁺/⁺*. Data shown as mean ± SD by two-way ANOVA, n=3 independent experiments for each group; *, p<0.01, **, p<0.001. (**B**) cAMP ELISA of undifferentiated bone marrow stromal cells (BMSCs), from 10 to 60 min with 100 nM PTH treatment, *Zfp467⁻/⁻* BMSCs had significantly higher level of intracellular cAMP after 10–60 min exposure of PTH. Data shown as mean ± SD by one-way ANOVA, n=3 independent experiments for each group; *, p<0.01, **, p<0.001. (**C, D**) Western blot and quantitative analysis of pre-differentiated COBs and BMSCs. Higher expression levels of p-CREB but not total CREB was found in both *Zfp467⁻/⁻* COBs and BMSC. Data shown as mean ± SD by unpaired Student's t test, n=3 independent experiments for each group. (**E, F**) Oxygen consumption rates (OCR) and extracellular acidification rates (ECAR) of undifferentiated *Zfp467⁺/⁺* and *Zfp467⁻/⁻* COBs or BMSCs. No difference was found regarding OCR between genotypes, but *Zfp467⁻/⁻* BMSCs had higher ECAR than *Zfp467⁺/⁺* BMSCs. Data shown as mean ± SD by unpaired Student's t test, n=12 technical replicates. *, p<0.01, **, p<0.001. (**G, H**) OCR and ECAR of *Zfp467⁺/⁺* and *Zfp467⁻/⁻* COBs or BMSCs after 3 days' osteogenic differentiation. No difference was found regarding OCR between genotypes, but both *Zfp467⁻/⁻* COBs and BMSCs cells had significantly higher ECAR level than *Zfp467⁺/⁺* cells. Data shown as mean ± SD by unpaired Student's t test, n=12 technical replicates. *, p<0.01, **, p<0.001.

The online version of this article includes the following source data for figure 7:

**Source data 1.** Western blot for *Figure 7C*.

**Source data 2.** Western blot for *Figure 7C* p-CREB in calvarial osteoblasts (COBs).

*Figure 7 continued on next page*

Figure 7 continued

Source data 3. Western blot for *Figure 7C* CREB in calvarial osteoblasts (COBs).

Source data 4. Western blot for *Figure 7C* ACTIN in calvarial osteoblasts (COBs).

Source data 5. Western blot for *Figure 7C* p-CREB in bone marrow stromal cells (BMSCs).

Source data 6. Western blot for *Figure 7C* CREB in bone marrow stromal cells (BMSCs).

Source data 7. Western blot for *Figure 7C* ACTIN in bone marrow stromal cells (BMSCs).

In order to prove that absence of *Zfp467* increases the expression of the PTH1R and influences PTH-induced osteoblastogenesis in vivo, *Zfp467* global KO mice was treated with PTH for 1 week and compared those responses to control mice. Although no much difference was observed between vehicle and the 1-week PTH-treated group in *Zfp467*⁺/⁺ mice, PTH-treated *Zfp467*⁻/⁻ mice had higher BT/TV (p=0.0572), Conn.D (p=0.0755) and lower Tb.Sp (p=0.0811) and SMI (p=0.0399) than PTH-treated *Zfp467*⁺/⁺ mice, even after only 1 week of treatment.

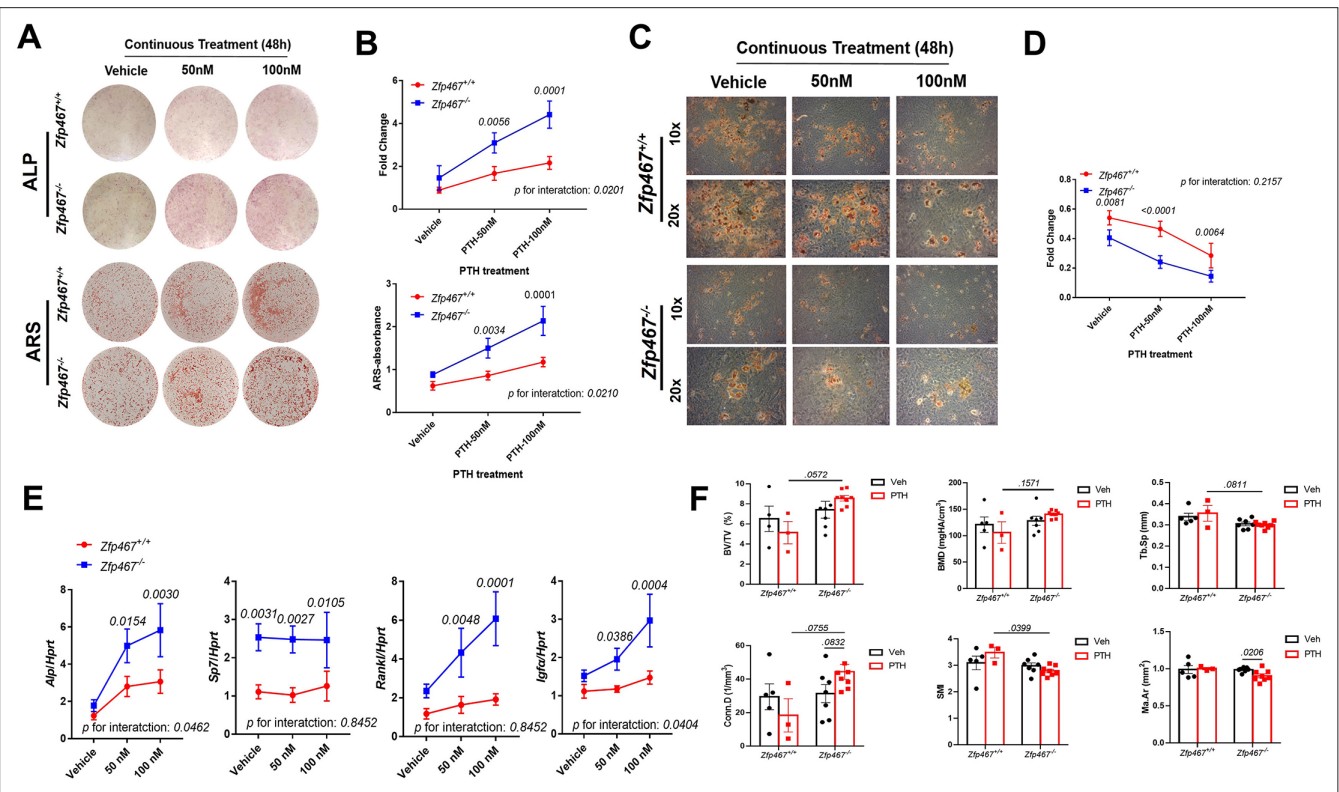

**Figure 8.** *Zfp467*⁻/⁻ cells showed increased sensitivity to parathyroid hormone (PTH) and enhanced pro-osteogenic as well as anti-adipogenic effects. (**A**) Representative images of alkaline phosphatase (ALP) (day 7) and alizarin red staining (ARS) (day 14) of differentiated calvarial osteoblasts (COBs) with PTH treatment. An increase in PTH dose led to an increase in ALP staining and mineralization in both *Zfp467*⁺/⁺ and *Zfp467*⁻/⁻ COBs. *Zfp467*⁻/⁻ COBs showed more ALP-positive cells and mineralization than *Zfp467*⁺/⁺ COBs. (**B**) ALP stain and ARS quantification in COBs. Data shown as mean ± SD by two-way ANOVA, n=3 independent experiments for each group. (**C**) Representative images of Oil Red O (ORO) staining (10× and 20×) of COBs after 14 days in' osteogenic differentiation with PTH treatment. While PTH treatment inhibited adipocyte formation in *Zfp467*⁺/⁺ and *Zfp467*⁻/⁻ groups, the *Zfp467*⁻/⁻ group showed fewer adipocytes in all treatment groups as compared to *Zfp467*⁺/⁺. (**D**) ORO stain quantification in COBs. Data shown as mean ± SD by two-way ANOVA, n=3 independent experiments for each group. (**E**) qPCR results for osteoblast-related genes after 7 days' osteogenic differentiation in COBs. Data shown as mean ± SD by two-way ANOVA, n=3 independent experiments for each group. (**F**) PTH treatment for female 12-week-old global *Zfp467*⁻/⁻ mice mice and control mice were measured using trabecular and cortical bone of tibiae after 1 week PTH treatment. Data shown as mean ± SD by two-way ANOVA, n=5–8 per group.

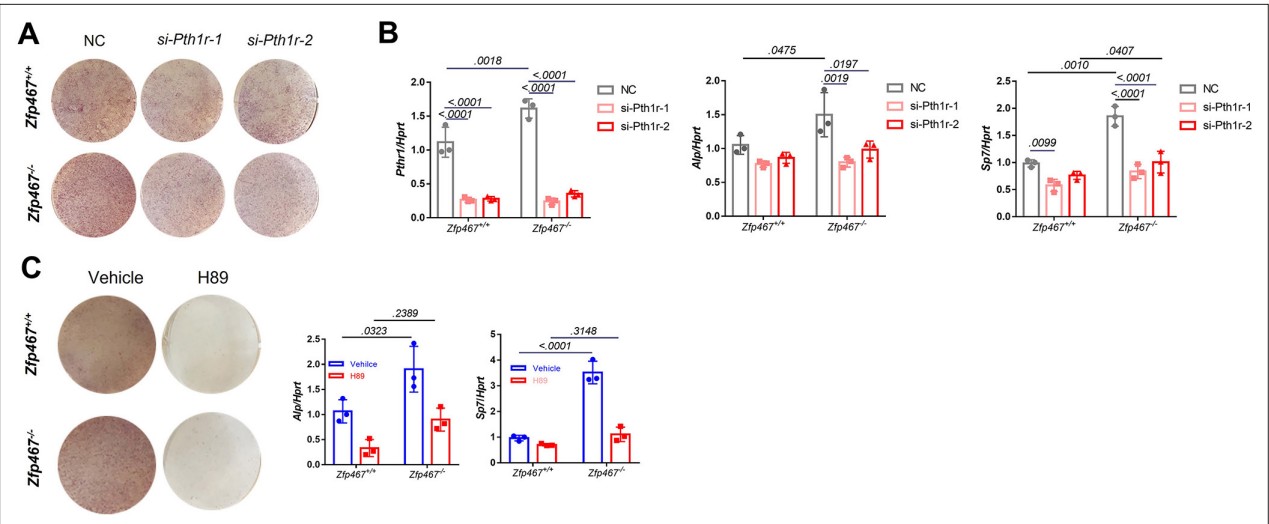

**Figure 9.** Gene silencing of *Pth1r* or PKA inhibitors suppressed *Zfp467⁻/⁻* induced increased osteogenic differentiation. (**A**) Representative images of alkaline phosphatase (ALP) staining of differentiated calvarial osteoblasts (COBs) with *Pth1r* or control siRNA treatment. (**B**) qPCR results for osteogenic differentiation-related genes after 7 days' osteogenic differentiation and siRNA treatment in COBs. Data shown as mean ± SD by two-way ANOVA, n=3 independent experiments for each group. (**C**) Representative images of ALP staining and qPCR results for osteogenic differentiation-related genes after 7 days' osteogenic differentiation and PKA inhibitor treatment in COBs. Data shown as mean ± SD by two-way ANOVA, n=3 independent experiments for each group. NC, negative control.

The online version of this article includes the following figure supplement(s) for figure 9:

**Figure supplement 1.** PKA inhibitor reversed the enhanced action of PTH on the osteogenic differentiation seen in Zfp467-/- COBs.

## Gene silencing of *Pth1r* or PKA inhibitors suppressed *Zfp467⁻/⁻* induced increased osteogenic differentiation

To determine whether the increase in osteogenic differentiation seen in *Zfp467⁻/⁻* cells is due to higher PTH1r levels, *Pth1r* was knocked down *by siRNA* and found that *Pth1r* knockdown led to decreased osteogenic differentiation in both *Zfp467⁺/⁺* and *Zfp467⁻/⁻* COBs. Although *Zfp467⁻/⁻* COBs still show slightly higher *Alp* staining (**Figure 9A**), *Pth1r* knockdown in *Zfp467⁻/⁻* cells dampens the increase in *Alp* and *Sp7* gene expression during osteogenic differentiation compared to *Zfp467⁺/⁺* cells, indicating that this increase is associated with the upregulation of *Pth1r* seen in *Zfp467⁻/⁻* cells (**Figure 9B**).

Furthermore, a PKA inhibitor could also significantly decrease osteogenic differentiation in COBs (**Figure 9C**). The staining showed ALP activity was much higher in *Zfp467⁻/⁻* cells in the absence of the PKA inhibitor, but no difference was found with the PKA inhibitor between genotypes (**Figure 9C**). In addition, with the PKA inhibitor, upregulated osteogenic genes in *Zfp467 ⁻/⁻* cells including *Alp* and *Sp7* could be totally reversed (**Figure 9C**). Similarly, treating COBs with the PKA inhibitor during PTH treatment simultaneously for 7 days led to a suppression of osteogenic differentiation in both *Zfp467⁺/⁺* and *Zfp467⁻/⁻* cells (**Figure 9—figure supplement 1**). qPCR results showed that PKA inhibitor could totally reverse the upregulation of *Alp*, *Sp7*, and *Rankl* in PTH-treated *Zfp467⁻/⁻* cells (**Figure 9—figure supplement 1A,B**).

## *Wdfy1*, *Sox10*, and *Ngfr* are downstream targets of ZFP467

Next question would be what were the downstream targets of Zfp467. In an unbiased analysis, using RNA-seq in pre- and post-differentiated COBs from *Zfp467⁺/⁺* and *Zfp467⁻/⁻* cells, potential regulatory pathways and differentially expressed genes (DEG) were obtained (**Figure 10A–D**). The PI 3K and MAPK signaling pathways were differentially upregulated in *Zfp467⁻/⁻* cells whether pre- or post-differentiated when compared to *Zfp467⁺/⁺* cells (**Figure 10D**). There were several highly expressed genes in the *Zfp467⁻/⁻* cells related to osteogenesis, including *Wdfy1*, *Sox10*, and *Ngfr* (**Figure 10B**). These results were confirmed by qRT-PCR (**Figure 10E**). When *Zfp467* was overexpressed in MC3T3-E1 cells those three genes were significantly suppressed relative to GFP overexpression.

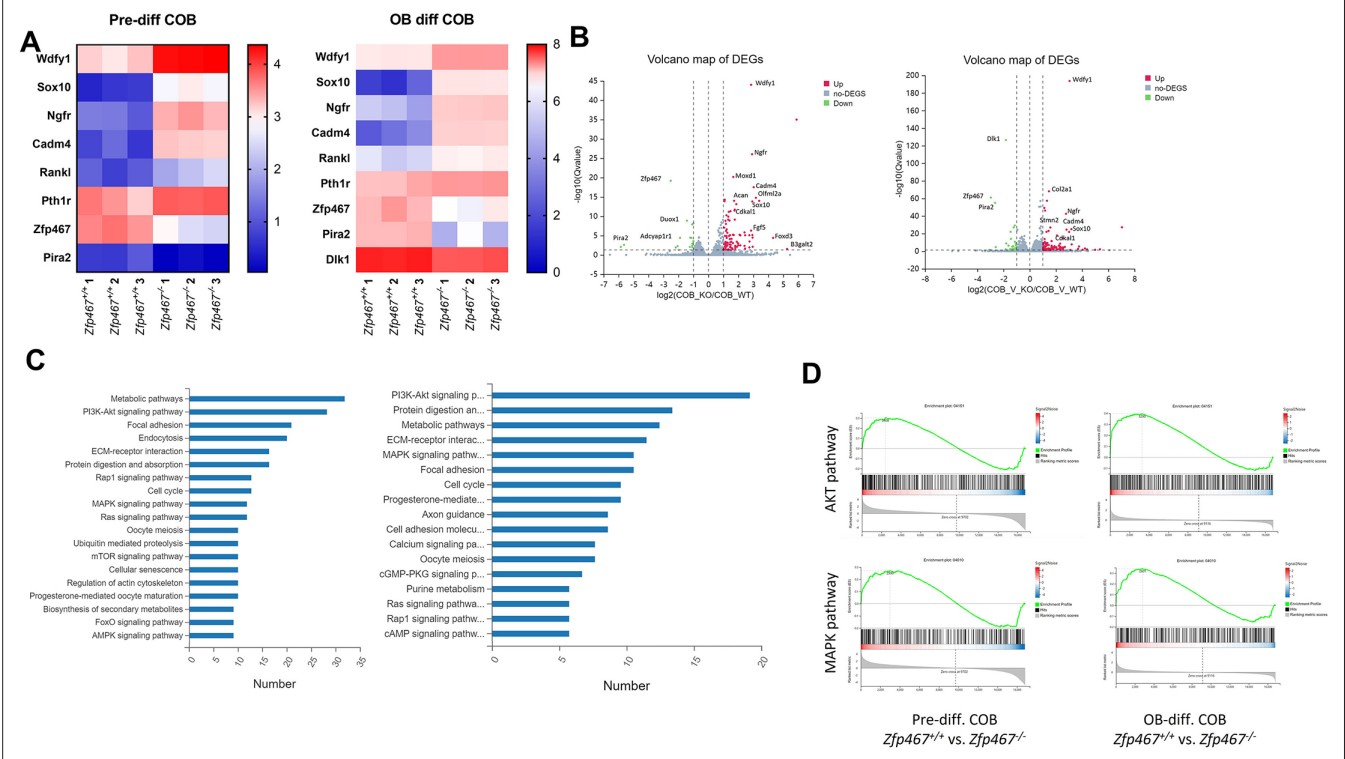

**Figure 10.** *Wdfy1*, *Sox10*, and *Ngfr* were found upregulated and MAPK, AKT pathways were found activated in *Zfp467⁻/⁻* calvarial osteoblasts (COBs). (**A**) Heat map of differentially expressed genes (DEG) from COBs (*Zfp467⁺/⁺* and *Zfp467⁻/⁻*) at pre-differentiation (Panel A) and at differentiation with a p-value <0.05 and a fold change >2.0 or <−2.0. (**B**) These DEGs that represented by volcano plots. (**C**) Functional annotation (Cellular Component [CC]) for the DEGs for *Zfp467⁺/⁺* vs *Zfp467⁻/⁻* for pre- (left) and post-differentiated COBs (right). (**D**) GSEA enrichment plots for AKT and MAPK pathways. (**E**) qRT-PCR was performed on COBs from *Zfp467⁺/⁺* and -/- to confirm gene expression changes noted by RNA-seq. (**F**) Overexpression of *Zfp467* in MC3T3-E1 cells confirmed statistically significant suppression of the top three genes (*Wdfy1*, *Sox10*, and *Ngfr*), when compared to GFP overexpression; p<0.05 or lower. Data shown as mean ± SD by unpaired Student's t test, n=5–8 independent experiments for each group.

## Discussion

Based on above data, the role of Zfp467 was identified in the molecular, cellular, and biochemical responses to PTH in osteoblast progenitors and in vivo. A novel pathway was reported by which PTH can enhance its own anabolic actions in osteoblast progenitors by repressing Zfp467, a negative regulator of PTH1R expression. *Zfp467* was originally isolated from mouse hematogenic endothelial LO cells as an OSM-inducible mRNA, which encodes protein ZFP467 (*Nakayama et al., 2002*), also called EZI. Zinc finger motifs are involved in protein-protein interactions and protein-DNA binding (*Ganss and Jheon, 2004*; *Leon and Roth, 2000*; *Berg, 1990*). Based on the zinc finger domain, ZFPs are classified into C2H2, C3H, and C4 (*Varnum et al., 1991*). ZFP467 belongs to C2H2 ZFP family whose zinc finger domain consists of two cysteine and two histidine residues. ZFP467 was initially found to cooperate with STAT3 and augment STAT3 activity by enhancing its nuclear translocation in a kidney cell line (*Nakayama et al., 2002*). ZFP467 is expressed ubiquitously and may be an important mediator of BMSC differentiation into the adipogenic or osteogenic lineages as well as functioning in other tissues in distinct ways (*Quach et al., 2011*; *You et al., 2012*; *You et al., 2015*).

Deletion of *Pth1r* in mesenchymal cells resulted in low bone mass, high marrow adiposity, and upregulation of *Zfp467* (*Fan et al., 2017*). In contrast, the absence of *Zfp467* resulted in a significant increase in Tb.BV/TV, accompanied by a marked reduction in peripheral and marrow adipose tissue and improved glucose tolerance (*Le et al., 2021*). Most importantly for the current study, whole bone marrow gene expression by qRT-PCR revealed an ~40% increase in *Pth1r* expression and greater protein levels in the *Zfp467⁻/⁻* mice compared to controls (*Le et al., 2021*). Therefore, PTH1R and ZFP467 could be involved in a feedback loop whereby the suppression of *Zfp467* mediated by PTH leads to an increase of PTH1R, and therefore an enhanced response to PTH treatment. However, due

to the global nature of the Zfp467 deletion, a cell non-autonomous effect could not be excluded. In the present study therefore, we tested that hypothesis both in vivo and in vitro, and sought to determine the cellular and biochemical mechanisms involved in this novel regulatory pathway.

First, we were able to show that conditional deletion of *Zfp467* by Prrx1Cre led to an increase in osteogenesis and bone formation, a finding that mirrors our results with the global deletion of Zfp467, although, it should be noted that the increased trabecular bone in *Prrx1Cre;Zfp467^{fl/fl}* was much more pronounced in male rather than female mice. On the other hand, using an AdipoqCre-driven system, no effect of conditional deletion in adipocytes on body composition, fat mass, bone mass, or total weight was observed. Taken together these data would suggest that *Zfp467* is an early mesenchymal transcriptional factor that regulates lineage allocation in early osteoblast but not adipocyte progenitors. Second, constitutive upregulation of *Zfp467* was observed when we genetically knocked down *Pth1r* in COBs and BMSCs. In addition, much like the study of *Quach et al., 2011*, acute PTH treatment could significantly suppress gene expression of *Zfp467* in both COBs and BMSCs, and the suppression could be partly rescued by both a PKA pathway inhibitor and a PKC inhibitor (*Quach et al., 2011*). Similarly, Forskolin, a PKA pathway activator, could also inhibit the expression of *Zfp467*, with a more sustained effect. These data indicated that PTH might suppress *Zfp467* expression via activation of the PTH1R through predominantly PKA pathways.

To test the validity of the hypothesis, how deletion of *Zfp467* affected the expression of the PTH1R was further investigated. The transcript of *Pth1r* that upregulated in *Zfp467* absent cells was further confirmed that via dual-fluorescence reporter assay that both P1 and P2 promoters of *Pth1r* were activated in *Zfp467* null cells; however, P2 was more activated than P1. Using three different transcription factor prediction databases including PROMO, JASPAR, and Animal TFDB, several candidate transcription factors that might be involved in the regulation of *Pth1r* in *Zfp467^{-/-}* cells via activation of the P2 promoter were observed. After overexpressing each candidate transcription factor, only NFκB1 could upregulate *Pth1r* via activation of the P2-2 Pth1r promoter. Further confocal immunofluorescence and nuclear protein detection indicated that the nuclear translocation of NFκB1 was much higher in *Zfp467^{-/-}* cells. Moreover, ChIP-qPCR results showed that NFκB1 could bind to the P2 promoter of the Pth1r. Taken together, these data suggested that the deletion of *Zfp467* resulted in higher nuclear translocation of NFκB1 which bound to the P2 promoter of Pth1r and promoted its transcription.

NFκB1 is one of the DNA binding subunits of the NF-kappa-B (NFκB) protein complex. NFκB is a transcriptional regulator that is activated by various stimuli including cytokines, bacteria, and oxidation. The NFκB pathway is involved in several biological processes including inflammation, bone resorption, aging, and cancer (*Cartwright and Perkins, 2016*). Activated NFκB1 translocates into the nucleus and stimulates the expression of an array of genes. However, no previous studies that NFκB1 may regulate the gene expression of *Pth1r* or other osteogenic-related genes. Nevertheless, it was reported that time- and stage-specific inhibition of endogenous inhibitors of IκB kinase (IKK)–nuclear factor-κB (NF-κB) in differentiated osteoblasts substantially increases trabecular bone mass and bone mineral density without affecting osteoclast activities (*Chang et al., 2009*). Therefore, activation of NF-κB may not only regulate osteoclast activation and bone resorption but also simultaneously be involved in osteoblast function, thus influencing bone formation and maintaining bone homeostasis. It is also conceivable that NFκB1 could associate with histone deacetylase-1 (HDAC1) or be regulated by a PKA catalytic subunit which is also downstream of PTH signaling in osteoblasts (*Yu et al., 2009*). Moreover, it is likely that other proteins like RelB that might bind to p50 and enhance its effect on the transcriptional regulation of *Pth1r* play a critical role in PTH-mediated osteogenesis. In addition to RelB-p50 heterodimers, more proximal events in the non-canonical NFκB signaling cascade was detected and higher phosphorylation level of both IKKα and p100 were observed in *Zfp467* knock down cells, which suggested the non-canonical NFκB pathway was activated when lacking *Zfp467*, and was involved in the transcriptional regulation of *Pth1r*.

Osteoblasts require substrates for energy utilization during collagen synthesis and mineralization. Previous reports by Lee et al. and Guntur et al. demonstrated that glycolysis is a major source of ATP for differentiating osteoblasts (*Maridas et al., 2019*; *Esen et al., 2015*). Consistently, *Zfp467^{-/-}* cells showed higher cAMP levels in response to PTH as well as higher glycolytic activity (i.e. ECAR). COBs from *Zfp467^{-/-}* mice also showed greater differentiation in osteogenic media but less differentiation into adipocytes compared to controls. Higher rates of osteogenesis, enhanced *Rankl* and *Igf1*

expression, increased glycolysis, and decreased adipogenesis are likely related to greater activation of the PTH1R, possibly due to its higher endogenous expression level. Consistently, after treating *Zfp467* global KO mice with PTH for 1 week and compared to control mice, PTH-treated *Zfp467*$^{-/-}$ mice showed higher bone volume and lower SMI than PTH-treated *Zfp467*$^{+/+}$ mice.

Since it was reported that the downstream effects of PTH could be partially blocked by PKA inhibitors (*Ishizuya et al., 1997*), *Pth1r* siRNA and PKA inhibitors were used to block the effect of PTH1R, which confirmed that PTH1R and its downstream pathways were involved in the deletion of *Zfp467*-induced osteogenic differentiation. ALP staining and qPCR results suggested that *Pth1r* siRNA and PKA inhibitor could nearly reverse the upregulated osteogenic differentiation in *Zfp467*$^{-/-}$ COBs with or without PTH treatment.

Last but not least, RNA-seq was performed to identify potential downstream targets and pathways of *Zfp467* that might be involved in the regulation of osteogenic differentiation. Among the top three upregulated genes in *Zfp467*$^{-/-}$ COBs, *Wdfy1* plays an important role in the innate immune responses by mediating TLR3/4 signaling (*Hu et al., 2015*; *Yang et al., 2020*), which might be also involved in the regulation of osteoblast metabolism and function (*Alonso-Pérez et al., 2018*; *Vega-Letter et al., 2016*). Another significantly upregulated gene, *Ngfr* in *Zfp467*$^{-/-}$ cells was found to be required for skeletal cell migration and osteogenesis, especially during injury repair (*Xu et al., 2022*; *Tomlinson et al., 2017*). NGFR could also mediate AMP-induced modulation of ERK1/2 and AKT cascades (*Piiper et al., 2002*), which is consistent with our GSEA pathways enrichment. However, further investigation is required to clarify how ZFP467 regulates *Wdfy1*, *Ngfr*, and their downstream AKT or MAPK pathways.

Despite uncovering a novel regulatory circuit through NFκB1, there are some limitations to our study. First, it should be noted that *Zfp467*$^{-/-}$ cells have higher nuclear translocation of GATA1 and overexpression of GATA1 could promote *Pth1r* expression. However, overexpression of GATA1 failed to activate the P2-2 promoter of *Pth1r*. Hence, that GATA1 might be another of the regulating transcription factors between ZFP467 and Pth1r, but it likely regulates *Pth1r* expression via binding sites other than those present on P2-2.

Second, how Zfp467 binds with NFκB1-RelB heterodimers was not determined, and we have not identified the exact genomic sequence required for NFκB1-RelB heterodimer binding at the P2 promoter binding site. Further studies will be needed to address this important mechanistic question. Hence the precise mechanism of action of Zfp467 and the downstream consequences of NFκB1-RelB heterodimer binding on the presumed switch between adipocytes and osteoblasts is still not fully defined. Last, to our knowledge, there must be a break on the activation of a feed-forward PTH-PTH1R regulatory system, but it is not delineated whereby the fast forward system interacts with β-arrestin or any other mechanism that turns off signaling.

In summary, taken together, we demonstrated the importance of the zinc finger protein ZFP467 for lineage allocation in vitro and in vivo, as well as responsiveness to PTH in osteoblast progenitors. Our data also support a novel feed-forward regulatory loop whereby suppression of *Zfp467* mediated by PTH and its downstream PKA pathway leads to an increase of PTH1R via NFκB1, and subsequent enhanced responsiveness to PTH treatment (*Figure 11*). In addition, our lines of evidence suggest that the NF-κB signaling pathway may be an important component of the osteoblastic response to cytokines, and possibly to factors elaborated by osteoclasts, enhancing coupling of resorption to formation. These findings have significant implications for our understanding of the anabolic effects of PTH on bone.

## Materials and methods

### Reagents

PTH were purchased from Bachem (Torrance, CA). H89, G06983 and Forskolin were purchased from Selleck Chemicals (Houston, TX). *Pth1r*, *Nfkb1*, *Zfp467*, and non-targeting control siRNAs were purchased from Life Technologies (Carlsbad, CA). The expression vector for EBF1, MYOD, cFOS, GATA1, NFκB1, and empty vector were obtained from Addgene: NFκB1 pcDNA3 was a gift from Stephen Smale (Addgene plasmid #20018), pcDNA3-GATA1 was a gift from Licio Collavin & Giannino Del Sal (Addgene plasmid #85693), pCAG-EBF1 was a gift from Elena Cattaneo (Addgene plasmid #96965), pCAG-MyoD was a gift from Andrew Lassar (Addgene plasmid #8398), MMTV-cFOS-SV40

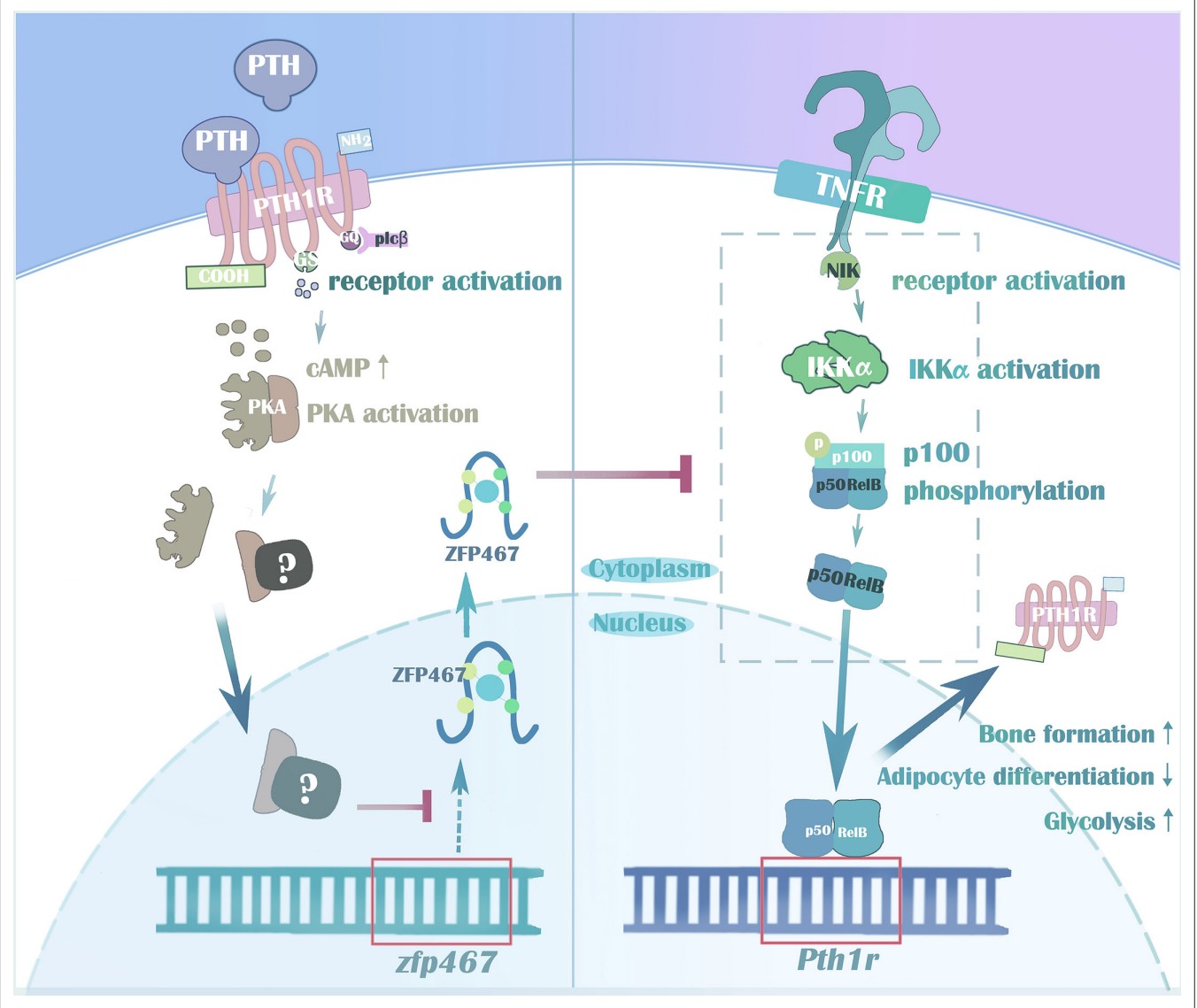

**Figure 11.** Parathyroid hormone (PTH) regulates osteogenesis and suppresses adipogenesis through Zfp467 in a feed-forward, PTH1R-cyclic AMP-dependent manner.

was a gift from Philip Leder (Addgene plasmid #19259). pCMV-Creb1 was a gift from Georgios Statho-poulos (Addgene plasmid # 154942).

MC3T3-E1 subclone 4 cell line was obtained from and authenticated by ATCC (CRL-2593) and was tested negative for mycoplasma.

## Animals

*Zfp467fl/fl* on the C57BL/6J background was generated by Cyagen (*Figure 1—figure supplement 1*). Exons 2–4 were selected as cKO region (Transcript: Zfp467-001 ENSMUST00000114561). The targeting vector, homology arms, and cKO region were generated by PCR using bacterial artificial chromosome clone RP24-144J8 and RP23-24K23 from the C57BL/6J library as template. To generate mice lacking Zfp467 in limb mesenchymal stem cells, *Prrx1Cre; Zfp467fl/fl* were generated by crossing *Prrx1Cre* transgenic mice to *Zfp467fl/fl* mice (*Prrx1Cre Zfp467fl/fl*) and *Zfp467.fl/fl* mice were used as controls. *AdipoqCre* mice on the C57BL/6J background was generated by Beth Israel Deaconess Medical Center. To generate mice lacking Zfp467 in adipocyte tissues, *AdipoqCre; Zfp467fl/fl* were generated by crossing *AdipoqCre* transgenic mice (C57BL/6J) to *Zfp467fl/fl* mice and *Zfp467 fl/fl* mice

were used as controls. Total DNA was isolated from ear punch biopsies, and routine PCR was used to genotype mice.

All experiments were performed with 12-week-old and sex-matched littermates. All animals are in the C57/Bl6 background and were housed in polycarbonate cages on sterilized paper bedding and maintained under 14:10 hr light:dark cycles in the barrier, AAALAC-accredited animal facility at Maine Medical Center Research Institute or Harvard Center for Comparative Medicine. All experimental procedures were approved by the Institutional Animal Care and Use Committee of Maine Medical Center (IACUC #1914) and followed the NIH guidelines for the Care and Use of Laboratory Animals and also approved by the Harvard University Institutional Animal Care and Use Committee.

No statistical methods were used to predetermine sample size. Mice label and measurements were performed by two independent researchers and selected at random (by cage) into following experiments. Masking was used during data collection and data analysis. Animals with ulcerative dermatitis or other diseases were excluded from the study.

## PTH treatment

PTH 1–34 bovine (PTH, Bachem, Torrance, CA) was made to 1 mg/mL stock using a vehicle solution (4 mM HCl contained 0.1% BSA).

In vivo study, at 12 weeks of age, both $Zfp467^{+/+}$ and $Zfp46^{-/-}$ female mice were subjected to either intraperitoneal PTH (Bachem, Torrance, CA, 80 µg/kg/day) or vehicle (Veh) injection daily for 1 week. Subsequently, mice were harvested and their tissues and serum were collected for analysis.

In vitro study, during osteogenic differentiation, PTH was administered by adding to osteogenic media every 48 hr for each group. For ELISA, PTH was administered by adding to culture media for 10–60 min.

## Dual-energy X-ray absorptiometry

Whole-body composition exclusive of the head was performed using the PIXImus densitometer (GE-Lunar, Fairfield, CT, USA). The PIXImus was calibrated daily with a phantom provided by the manufacturer.

## Micro-computed tomography

A high-resolution desktop micro-tomographic system (vivaCT 40, Scanco Medical AG, Brüttisellen, Switzerland) was used to assess the trabecular and cortical bone microarchitecture, volume, and mineral density in mouse femurs. Scans were acquired using a 10.5 µm³ isotropic voxel size, 70 kVp peak X-ray tube intensity, 114 mA X-ray tube current, 250 ms integration time, and were subjected to Gaussian filtration and segmentation. All scans were analyzed using manufacturer software (Scanco, version 4.05). The trabecular bone region of interest started 210 µm (20 transverse slices) proximal to the break in the growth plate and extended 1575 µm (150 transverse slices). Bone was segmented from soft tissue using a mineral density threshold of 375 mg HA/cm³. Trabecular bone was analyzed for Tb.BV/TV (%), Tb.Th (mm), Tb.N (mm⁻¹), Tb.Sp (mm), Conn.D (mm⁻³), and trabecular bone mineral density (Tb.BMD, mg HA/cm³). The cortical bone region of interest started at 55% of the total bone length distal to the femoral head and extended 525 µm (50 transverse slices). Bone was segmented using a mineral density threshold of 700 mg HA/cm³. Cortical bone was analyzed for bone area (Ct.Ar, mm²), medullary area (Ma.Ar, mm²), bone area fraction (Ct.Ar/Tt.Ar, %), cortical thickness (Ct.Th, µm), cortical tissue mineral density (Ct.TMD, mg HA/cm³).

## Marrow adipose tissue quantification by osmium tetroxide staining and µCT

At the time of sacrifice, tibiae were isolated and placed into 10% neutral buffered formalin overnight at 4°C. Soft tissue was carefully removed to ensure that the fibula remained intact and the bones were washed under continuous cold PBS for 1 hr, then stored in PBS at 4°C. Quantification and visualization of marrow adipose tissue was performed as described previously (*Scheller et al., 2014*). Briefly, bones were decalcified in 14% EDTA (pH 7.4) for 14 days, with EDTA changes every 3–4 days. Bones were then washed for 10 min in PBS (three times) and stained with a 1:1 mixture of 2% aqueous osmium tetroxide (cat# 23310-10, Polysciences, Inc, Warrington, PA, USA) and 5% potassium dichromate for 48 hr. Stained bones were then washed with PBS (pH 7.4) for 5 hr (three times), and subsequently

scanned by μCT. BMAT content was calculated by determining the whole volume of second osseous center of tibiae.

## Bone histomorphometry

Mice were intraperitoneally injected at 9 and 2 days, respectively, prior to sacrifice, with 20 mg/kg calcein and 30 mg/kg alizarin complexones (Sigma, St. Louis, MO). Static and dynamic histomorphometry measurements were performed in *Prrx1Cre; Zfp467^{fl/fl}* and *Zfp467^{fl/fl}* mice at 12 weeks of age as previously described (*Le et al., 2021*). Tibiae were analyzed as described and standard nomenclature was used.

## Primary cells isolation

The generation of *Zfp467^{-/-}* and wild-type mice was previously described. COBs were isolated from calvarias of 3- to 5-day-old *Zfp467^{+/+}* and *Zfp467^{-/-}* neonates as described in the following protocols (*Kawai et al., 2011*; *Rosen et al., 1997*). BMSCs were isolated from tibiae and femurs of 6-week-old *Zfp467^{+/+}* and *Zfp467^{-/-}* female mice as described in the previously established protocols (*Le et al., 2021*; *Maridas et al., 2018*). All studies were reviewed and approved by the Institutional Animal Care and Use Committee of Maine Medical Center and followed the NIH guidelines for the Care and Use of Laboratory Animals.

## RNA interference and plasmid transfection

Twenty-four hr after seeding, RNA oligos were transfected into COBs or BMSCs using Lipofectamine RNAiMAX Transfection Reagent according to the manufacturer's instructions for 48 hr and the final concentration of siRNA was 5 nM. ORF clone expression vector or a controlled vector was introduced into COBs, BMSCs, or MC3T3-E1 cells using Lipofectamine 3000 Transfection Reagent for 72 hr once cells reached 80% confluence. All transfection reagents were purchased from Thermo Fisher Scientific, Waltham, MA.

## Real-time PCR and western blot

Total RNA was isolated using a standard TRIzol extraction (Life Technologies, Carlsbad, CA) method. cDNA was generated using the High Capacity cDNA Reverse Transcription Kit (Life Technologies, Carlsbad, CA) according to the manufacturer's instructions for real-time PCR.

Proteins from cell culture were extracted by scraping the culture wells in the presence of RIPA buffer (Bio-Rad, Hercules, CA) with protease inhibitor and phosphatase inhibitor (St. Louis, MO). Cytoplasm and nuclear protein were extracted using Nuclear and Cytoplasmic Extraction Reagent Kit (Thermo Scientific, MA, USA). Antibodies used for western blot and ChIP or co-IP were listed in *Table 2*.

**Table 2.** Antibody list for co-IP, ChIP, and western blot.

| Antibody | Supplier | Cat Num |
| --- | --- | --- |
| ACTIN | Santa Cruz Bio | SC47778 |
| PTH1R | Sigma | SAB4502493 |
| GATA1 | Cell Signaling Technology | 3535T |
| NF$\kappa$B1 (IP) | Cell Signaling Technology | 13586S |
| NF$\kappa$B1 (IB) | Cell Signaling Technology | 13681S |
| Histone H3 | Cell Signaling Technology | 9715S |
| RelB | Santa Cruz Bio | sc-48366 |
| NPAS1 | Santa Cruz Bio | sc-376083 |
| p-CREB (SER133) | Cell Signaling Technology | 9198S |
| CREB | Cell Signaling Technology | 9197T |

**Table 3.** Primer list of ChIP-qPCR for P2-2 promoter of *Pth1r*.

| Prime sequence | |
| --- | --- |
| P2-2-Forward1 | CCATCTCTCTCACTTTCCCCAAG |
| P2-2-Reverse1 | ATCCCTGGTTCTTCGATCTAGCCC |
| P2-2- Forward2 | CCTAGCTGAACCCGAGTCTTG |
| P2-2- Reverse2 | GTCTAGCGGATCGGAGACTCT |
| P2-2- Forward3 | AACCGGGAGTCCAACGAAGGT |
| P2-2- Reverse3 | GGTCTGGCTATGTGGGGAC |
| P2-2- Forward4 | GGCTGCATAGCCTGGTTCTAGC |
| P2-2- Reverse4 | CCCACTACCCCGATCTTCCGG |
| P2-2- Forward5 | ACGGCGCGAGAAATACCAGGAG |
| P2-2- Reverse5 | CGTGGCTGGGACGTTGTCTC |

## Nuclear translocation detection

*Zfp467*[+/+] and *Zfp467*[-/-] BMSCs were plated on an eight-chamber slide (Sigma, St. Louis, MO). Protein location of GATA1 and NFκB1 was detected by immunofluorescence using Confocal Microscopy (Leica DMI6000). Nuclear protein was isolated from baseline *Zfp467*[+/+] and *Zfp467*[-/-] BMSCs lysate by Pierce NE-PER Nuclear and Cytoplasmic Extraction Reagent Kit (Life Technologies, Carlsbad, CA). Protein level of GATA1 and NFκB1 was measured using nuclear extracts. Related quantification was performed by ImageJ software.

## Dual-fluorescence reporter assay

The Pth1r P1 (0.5 kb: 0–500, 1.1 kb: –600 to 500; 1.5 kb: –1109 to 500, 2.1 kb: –1598 to 500, P1: –581 to –1109) and P2 promoters (P2-1: –449 to 0, P2-2: –212 to –826) were cloned through PCR into the pGL4.20 luciferase reporter vector (Promega, Madison, WI) using C57BL/6 genomic DNA as a template. pGL4.75 luciferase reporter was used as a positive control for transfection efficiency normalization. Baseline COBs, BMSCs, or MC3T3-E1 cells were transfected with the reporter constructs and incubated for 48 hr. Luciferase activity was measured using GloMax-20/20 (Promega, Madison, WI). The transcriptional activity was expressed as the ratio of firefly:Renilla luciferase activity.

## ChIP assay and biotin-*Pth1r* pulldown

MC3T3-E1 subclone 4 (CRL-2593) cell line was purchased from ATCC and used for ChIP and DNA pull-down assay. When MC3T3-E1 cells reached confluency, they were then fixed in 1% paraformaldehyde/PBS for 10 min at room temperature. Chromatin shearing and immunoprecipitation were performed using EZ-Magna ChIP A/G Chromatin Immunoprecipitation Kit (Sigma, St. Louis, MO) according to the manufacturer's instructions. The immunoprecipitated DNA fragments were used as templates for PCR amplification. ChIP-PCR primers sequences were listed in *Table 3*. PCR products were used for nucleic acid electrophoresis to avoid unspecific amplification.

DNA pulldown assay was performed according to previously reported protocol (*Jutras et al., 2012*). 5′ biotin-modified Pth1r P2-2 dsDNA probe was generated using PCR with oligonucleotide primers modified at its 5′ end by Sangon Biotech (Shanghai, China). Nuclear protein was extracted using Nuclear and Cytoplasmic Extraction Reagent Kit (Thermo Scientific, MA, USA). NFκB1 purified protein was purchased from Proteintech (Rosemont, USA). Briefly, the Pth1r P2-2 or oligo probe was incubated with streptavidin-coupled Dynabeads (Invitrogen, USA) at room temperature for 1 hr to generate probe-bound Dynabeads, and then the probe-bound Dynabeads were incubated with MC3T3-E1 nuclear extracts or NFκB1 purified protein at 4°C overnight. The protein bound to the probe and beads were eluted and used for gel electrophoresis.

## Determination of cellular cAMP levels

Intracellular cAMP expression levels were measured after 10 min to 48 hr of PTH treatment using Mouse/Rat cAMP Assay Parameter Kit (R&D Systems, Minneapolis, MN) according to the manufacturer's instructions in both pre-differentiation COBs and BMSCs. Total protein quantity was used for normalization using the Pierce BCA Protein Assay Kit (Life Technologies, Carlsbad, CA).

## Cellular respiration measurements

The OCR and ECAR were assessed pre-differentiation or 3 days after osteogenic differentiation using the XF96 Extracellular Flux Analyzer (Seahorse Biosciences, North Billerica, MA), as previously described (*Maridas et al., 2018*). Briefly, BMSCs and COBs were plated in 96-well Seahorse XF96 Cell Culture Microplates (Seahorse Biosciences, North Billerica, MA). Several mitochondrial electron transport chain complex inhibitors were used during the test: 2.52 µM oligomycin, 12.65 µM carbonyl-cyanide *p*-(triflueormethoxy) phenylhydrazone, and 12.67 µM rotenone. All inhibitors were purchased from Seahorse Biosciences. Cell counts via fluorescence detection with Hoechst (Life Technologies, Carlsbad, CA) were used for normalization.

## Osteogenic differentiation and related measurements

In vitro osteoblast differentiation and measurement were done according to previously published protocols (*Le et al., 2021*). Briefly, BMSCs and COBs were plated at a density of $1\times10^6$/well in six-well plates. When cells reached around 80% confluency, osteogenesis induction began using osteogenic induction media which consisted of complete αMEM (αMEM, 5% fetal bovine serum, and 1% penicillin/streptomycin), 50 µg/mL ascorbic acid, and 8 mM beta-glycerophosphate (both were purchased from Sigma, St. Louis, MO). αMEM and penicillin/streptomycin were purchased from Life Technologies (Carlsbad, CA), FBS was purchased from VWR (Radmor, PA). Medium was changed every other day until cells were ready to stain for ALP and ARS to assess osteoblasts and mineralization, respectively, around day 7 and day 14 after differentiation. Additionally, RNA was extracted on day 7 after differentiation for real-time PCR analysis.

## ALP and ARS

ALP staining was performed using ALP kit obtained from Sigma (St. Louis, MO) according to the manufacturer's instructions at day 4 and day 7 after osteogenic differentiation for BMSCs and COBs, respectively. ARS (Sigma, St. Louis) staining was done using 1% solution at pH 4.2 at day 14 after osteogenic differentiation for both BMSCs and COBs. Briefly, after fixation, cells were stained for 30 min with ARS solution at room temperature. Cells were then washed a couple of times with water before they were visualized under the microscope (Leica DM IRB, TV camera). Five random fields were chosen to capture images using the ZEISS Efficient Navigation, blue edition camera (Bloomfield, CT) per treatment group for quantification. Alizarin red-stained cells were destained with 10% cetylpyridinium chloride for 1 hr, and their absorbance was measured at 570 nm using a plate reader (MRX Dynex Technologies, Chantilly, VA) for quantitation.

## Oil Red O staining

Differentiated adipocytes were stained using Oil Red O (ORO) solution at day 9 after adipogenic diff. Briefly, cells were fixed with 10% neutral buffer formalin (Sigma, St. Louis). After that, they were washed with 60% isopropanol (Sigma, St. Louis, MO) before stained with ORO solution (Sigma, St. Louis, MO) for 15 min at room temperature. Cells were then washed a couple of times with water before they were visualized and pictures were taking using the Zeiss microscope. For quantification, lipid from adipocyte droplets was extracted using isopropanol and the absorbance was measured at 490 nm using a plate reader (MRX Dynex Technologies, Chantilly, VA).

## RNA-seq and analysis

RNA-seq was performed on samples from COBs isolated from *Zfp467*[+/+] and *Zfp467*[-/-] mice, pre- or post-4 days' osteogenic differentiation. RNA-seq analysis was performed by BGI Group (UK, China). Samples were de-identified, and the analysis was blinded to assignment. We employed DAVID 6.8 to perform the functional annotation (Cellular Component [CC]) and GSEA enrichment plots for related pathways. Protein coding genes were assessed with p<0.05, fold change >2.0 or <−2.0. The enriched

CC was evaluated using a false discovery rate (Benjamini–Hochberg method) of 0.1 and DEG number. We have uploaded our original sequencing data to Sequence Read Archive database (PRJNA877934, http://www.ncbi.nlm.nih.gov/bioproject/877934).

## Statistical analysis

All data are expressed as the mean ± standard deviation (SD) unless otherwise noted. Results were analyzed for statistical differences using Student's t test between two groups or two-way ANOVA followed by Bonferroni's multiple comparison post hoc test among three or more groups where appropriate. All statistics were performed with Prism GraphPad 7.0 statistical software (GraphPad Software, Inc, La Jolla, CA). Values of $p<0.05$ were considered statistically different.

## Acknowledgements

We acknowledge the intellectual contribution and mice from Dr. Beate Lanske at Discovery Sciences. This study was supported by NIDDK R01 DK112374-RB and CJR; NIAMS R01AR073774- RB and CJR; NIDDK R24 DK092759-CJR.

## Additional information

### Funding

| Funder | Grant reference number | Author |
| --- | --- | --- |
| National Institute of Diabetes and Digestive and Kidney Diseases | DK112374 | Clifford J Rosen |
| National Institute of Arthritis and Musculoskeletal and Skin Diseases | AR073774 | Clifford J Rosen |
| National Institute of Diabetes and Digestive and Kidney Diseases | DK092759 | Clifford J Rosen |

The funders had no role in study design, data collection and interpretation, or the decision to submit the work for publication.

### Author contributions

Hanghang Liu, Conceptualization, Data curation, Software, Formal analysis, Supervision, Funding acquisition, Investigation, Methodology, Writing - original draft, Project administration, Writing – review and editing; Akane Wada, Data curation, Software, Formal analysis, Investigation, Visualization, Methodology, Writing - original draft; Isabella Le, Software, Formal analysis, Validation, Investigation, Visualization, Methodology; Phuong T Le, Data curation, Software, Formal analysis, Validation, Investigation, Methodology, Project administration, Writing – review and editing; Andrew WF Lee, Validation, Investigation; Jun Zhou, Data curation, Visualization, Methodology, Project administration, Writing – review and editing; Francesca Gori, Conceptualization, Supervision, Visualization, Methodology, Writing – review and editing; Roland Baron, Conceptualization, Supervision, Funding acquisition, Writing – review and editing; Clifford J Rosen, Conceptualization, Supervision, Funding acquisition, Project administration, Writing – review and editing

### Author ORCIDs

Francesca Gori http://orcid.org/0000-0001-5685-8303
Clifford J Rosen http://orcid.org/0000-0003-3436-8199

### Ethics

All experimental procedures were approved by the Institutional Animal Care and Use Committee of Maine Medical Center (IACUC # 1914) and followed the NIH guidelines for the Care and Use of Laboratory Animals and also approved by the Harvard University Institutional Animal Care and Use Committee.

Decision letter and Author response
Decision letter https://doi.org/10.7554/eLife.83345.sa1
Author response https://doi.org/10.7554/eLife.83345.sa2

## Additional files

### Supplementary files
• Transparent reporting form

### Data availability

RNA-seq data that support the findings in this study are openly available in Sequence Read Archive database (PRJNA877934, http://www.ncbi.nlm.nih.gov/bioproject/877934).All data generated or analysed during this study are included in the manuscript and supporting file; Source Data files have been provided for Figures 3-Figure 7.

The following dataset was generated:

| Author(s) | Year | Dataset title | Dataset URL | Database and Identifier |
|---|---|---|---|---|
| Rosen C | 2022 | PTH regulates osteogenesis and suppresses adipogenesis through Zfp 467 in a feed-forward cyclic AMP dependent manner | https://www.ncbi.nlm.nih.gov/bioproject/PRJNA877934 | NCBI BioProject, PRJNA877934 |

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
