## [Editor Report]

The study provides evidence that the hormone PTH increases bone mass by, at least in part, regulating the factor Zfp467. In turn, Zfp67 controls expression of the receptor for PTH, thus creating a feedback loop that overall augments bone mass. The findings are novel and of great interest. The study is significant as it unveils a novel feedback loop involving PTH, a critical endocrine regulator of calcium, phosphate, and bone mass.

---

## [Decision Letter]

**Decision letter after peer review:**

Thank you for submitting your article "PTH regulates osteogenesis and suppresses adipogenesis through Zfp 467 in a feed-forward, cyclic AMP-dependent manner" for consideration by *eLife*. Your article has been reviewed by 3 peer reviewers, and the evaluation has been overseen by a Reviewing Editor and Carlos Isales as the Senior Editor. The following individual involved in review of your submission has agreed to reveal their identity: Matthew B Greenblatt (Reviewer #1).

Essential revisions:

1) To study more in depth potential mechanisms downstream of Zfp467.

2) To provide more convincing evidence that the pathway as described in the study is relevant in vivo.

*Reviewer #1 (Recommendations for the authors):*

1. How does Zfp467 regulate p50? This is an important step in the molecular pathway laid out in this manuscript that doesn't appear to be fully established. Given that post-translational regulation is essential for NF-κB activity, further exploration of whether or not more proximal events in the NF-κB signaling cascade are impacted in Zfp467-deficient cells, such as p105 phosphorylation and proteasomal processing to p50, is important to fully flesh out this aspect of the phenotype. Additionally, it is not totally clear from the provided immunoblots and immunofluorescence studies that p50 actually displays increased activation in Zfp467-deficient cells, as the difference is subtle. What NF-ΚB-activating stimulus is proposed to be upstream of the activation seen here or more generally for the Zfp467 regulatory network studied here? Conducting similar studies in the presence of explicit NF-κB stimulation may be helpful to clarify the p50 activation phenotype.

2. It is a little bit confusing to follow due to the absence of figure labels (see the comment below) but it seems like the data on Prx1-cre Zfp467 fl/fl mice is missing from Figure 1? What appears to be Figure 1 only has data on the adipoQ-cre cross, not Prx1-cre mice. For any areas where important uCT phenotypes are noted, providing 3D reconstructions (ideally of an entire metaphyseal/epiphyseal region) is recommended to help visually communicate any architectural differences.

*Reviewer #2 (Recommendations for the authors):*

The authors have analyzed the effect of the zinc finger transcription factor, ZFP467, on PTH signaling in the osteoblast and vice-versa. They have established that in the absence of osteoblastic ZFP467, there is an increase in trabecular bone, osteoblast differentiation and PTH signaling and PTH1R. PTH, in turn, transiently decreases gene expression of Zfp467. They analyzed the promoters of the Pth1r gene and found a region upstream of the P2 promoter that seems to be most influenced by the absence of ZFP467. They concluded that p50 of the NFKB complex binds to this promoter and is more active in the absence of ZFP467. They have established a regulatory feedback mechanism of this transcription factor and PTH. However, they have not established the actual mechanism of action of the transcription factor nor which gene it acts on in the osteoblast. There are a number of omissions in the manuscript that would improve it, as well as other experiments that should be conducted to make it more compelling. At present, it is descriptive and does not address the central issue of how ZFP467 acts.

Specific Comments:

1. The overall manuscript has been prepared poorly. It required several emails to obtain the correct figures for Figure 1 and Figure 1-supplement 2. As well, there are a number of errors in the figure legends and Materials and methods. For instance, in Figure legends 1 and supplement 2, the legends state that there are representative uCT images (although the English here is poor), but these are not provided. There are abbreviations that are not defined, e.g., U and TSS in Figure 3 legend, NC in Figures 5, 6 and 9. In the Materials and methods, the DXA procedure "of the head" does not make sense. The PTH treatment is given as 48 h, when they have time courses of 10-60 min. The entire manuscript needs careful review by the senior authors.

2. The Prrx1Cre/Zfp467 mice do not completely recapitulate the global knockout. They have reduced cortical thickness in the females, and no significant change in the males, opposite to the global knockouts. They say the global knockouts had increased cortical thickness, but it also was not significant, and the sexes were not given in that paper. Thus, they need to modify the language in the present manuscript. They also need to modify the conclusion of this section to "maybe attributable to changes in the MSC lineage".

3. The doses of PTH they have used with the cultured cells are very high, 5 x 10-8 M and 10-7 M. These doses often inhibit growth of osteoblastic cells. They never provide a dose response of PTH action, and there is no difference in effect between the two doses. In Figure 2., the decrease in Zfp467 mRNA expression is very transient, only observed at 10 min and returning to control by 30 min. They must provide protein levels of ZFP467 to establish that the gene expression is reflected in change in the protein, otherwise, it is not relevant.

4. In Figure 3, they should provide the relative levels of the three transcripts to each other, rather than using fold expression to the +/+ cells. Also, the luciferase data should be in units, not fold change to the NC (don't know what this is, "no transfection" or "no construct"?). Here, it is very clear that P1 is a poor promoter. The data here should also be tested as an ANOVA since they are using the NC to give fold change.

5. Figure 4., the overexpression data should be judged by Westerns. The nuclear immunofluorescence expression of GATA1 and p50 should be quantitated.

6. Figure 4., supplement, it is not apparent whether the data are fold change to one of the samples, or relative expression. If the former, the sample seems to vary.

7. Why don't they perform ChIP-seq for ZFP467? This would give them much more direct information as to how this transcription factor functions. They do not seem to have evidence that it binds to the promoters of the Pthr1 gene, but ChIP-seq may give them that.

8. Figure 5C., they need to provide more information as to what each of these regions of p2 are, base pairs etc. Figure 5F., the PTHR1 data need to be quantitated.

9. Figure 6D. These immunoprecipitations are poor. There is no band in the input for RelB or NPAS1, and the quality of the blots is unsatisfactory.

10. Figure 7A and B, the data should be presented as time course lines, not histograms. Panel C does not provide much information, except to say that there may be higher baseline cAMP in the -/- cells. They need to perform PTH treatment of the cells and determine if there is a difference in pCREB. Panels E-G are not particularly meaningful, and it is not apparent how they connect to the function of ZFP467.

11. Figure 8, they cannot call use of two high doses of PTH a dose response. In panel A, they can elute the alizarin red and obtain quantitation. What is 1 in panel D? In panel E, there is no PTH effect on Sp7, nor in the +/+ cells for Rankl (which is very surprising for COBs), and these exceptions need to be noted in the text.

12. The RNAseq of the two cell populations is disappointing since there do not appear to be any osteoblastic genes, but a number of chondrocyte genes changed. In addition, they note that the PI-3-K and MAPK pathways are differentially upregulated in the -/- cells, but these were never investigated.

*Reviewer #3 (Recommendations for the authors):*

First let me start by reporting that there is discrepancy in the figures depending on the version that is downloaded from the submission files. In the PDF version of the paper where the figures are incorporated in the text there is a graphical abstract (never mentioned in the text) and the rest of the Figures correspond to the text except for supplementary Figure 2 where some of the microCT (relative to cortical bone) data are missing. In the PDF version where the Figures are placed at the end of the document, there is no graphical abstract, Figure 1 is missing (it can only be seen in the other version and it is very small). In addition, there are 2 Figures relative to the phenotype of the AdipoCre Zfp467 mice. The second figure, which contains BMD and microCT (cortical data), is not mentioned in the Figure legend. Moreover, in Supplementary Figure 5, the panels are different depending on the version that is downloaded.

Labelling each figure would be very helpful as well as calling the Supplemental Figures only supplemental figures and not Figure x—figure supplemental.

1) The authors claim that PrrxCre Zfp467mice have an identical phenotype compared to Zfp467 global knock out mice (reference 14). However, Zfp467 global KO mice have increased cortical thickness (females) whereas in this paper the females have decreased cortical thickness. This discrepancy need to be explained and discussed. Moreover, the adipose tissue volume fraction is decreased in Zfp467 global knock out females but is not decreased in PrrxCre Zfp467 females. Overall, the trabecular phenotype is much less prominent and does not "perfectly recapitulate" the phenotype of the KO mice, therefore the authors cannot conclude that the changes in the Zfp467 KO mice are mostly attributable to the lack of Zfp467 in mesenchymal progenitors.

2) In view of the differences in the phenotype between PrrxCre Zfp467 mice and global KO mice, to claim that Zfp467 KO mice have increased osteogenic differentiation and bone formation, histomorphometric data showing increased Osteoblast number or BFR should be reported.

3) Although the in vitro data are convincing and demonstrate that lack of Zfp467 increases the expression of the PTH1R and influences PTH-induced osteoblastogenesis, a clear proof of this mechanism would be treatment of PrrxCre Zfp467 and control mice with PTH to assess the difference in the magnitude of the response to PTH in vivo.

---

## [Author Response]

Reviewer #1 (Recommendations for the authors):1. How does Zfp467 regulate p50? This is an important step in the molecular pathway laid out in this manuscript that doesn't appear to be fully established. Given that post-translational regulation is essential for NF-κB activity, further exploration of whether or not more proximal events in the NF-κB signaling cascade are impacted in Zfp467-deficient cells, such as p105 phosphorylation and proteasomal processing to p50, is important to fully flesh out this aspect of the phenotype. Additionally, it is not totally clear from the provided immunoblots and immunofluorescence studies that p50 actually displays increased activation in Zfp467-deficient cells, as the difference is subtle. What NF-ΚB-activating stimulus is proposed to be upstream of the activation seen here or more generally for the Zfp467 regulatory network studied here? Conducting similar studies in the presence of explicit NF-κB stimulation may be helpful to clarify the p50 activation phenotype.

Thank you for your suggestions. We measured p50, p105 and p105 phosphorylation in Zfp467 siRNA treated MC3T3-E1 cells, and we did not find any difference between groups (Figure 6E). Therefore, Zfp467 may have no effect on p105 proteasomal processing and p50 production.

We agree that it is possible there is a signaling pathway between zfp467 and the NF-κB non-canonical network. So we detected the expression level of few classic non-canonical NFKB pathway stimuli and related receptors in Zfp467 siRNA treated cells and found that Rankl was significantly upregulated, which is also in consistent with our RNA-seq data (Figure 10A). Interestingly, when treating MC3T3-E1 cells with 10uM RANKL for 24 hours, transcription levels of Pth1r were significantly upregulated. (Figure 6—figure supplement 1). We are planning to follow up on these observations regarding RANKL regulation of PTH1R.

2. It is a little bit confusing to follow due to the absence of figure labels (see the comment below) but it seems like the data on Prx1-cre Zfp467 fl/fl mice is missing from Figure 1? What appears to be Figure 1 only has data on the adipoQ-cre cross, not Prx1-cre mice. For any areas where important uCT phenotypes are noted, providing 3D reconstructions (ideally of an entire metaphyseal/epiphyseal region) is recommended to help visually communicate any architectural differences.

We apologize for the mistake in labeling so we re-updated Figure. 1 with histomorphometry data for Prx1-cre mice. We also now provide uCT images reconstructed 3D for the Prrx1CreZfp467 mice, both males and females.

Reviewer #2 (Recommendations for the authors):The authors have analyzed the effect of the zinc finger transcription factor, ZFP467, on PTH signaling in the osteoblast and vice-versa. They have established that in the absence of osteoblastic ZFP467, there is an increase in trabecular bone, osteoblast differentiation and PTH signaling and PTH1R. PTH, in turn, transiently decreases gene expression of Zfp467. They analyzed the promoters of the Pth1r gene and found a region upstream of the P2 promoter that seems to be most influenced by the absence of ZFP467. They concluded that p50 of the NFKB complex binds to this promoter and is more active in the absence of ZFP467. They have established a regulatory feedback mechanism of this transcription factor and PTH. However, they have not established the actual mechanism of action of the transcription factor nor which gene it acts on in the osteoblast. There are a number of omissions in the manuscript that would improve it, as well as other experiments that should be conducted to make it more compelling. At present, it is descriptive and does not address the central issue of how ZFP467 acts.Specific Comments:1. The overall manuscript has been prepared poorly. It required several emails to obtain the correct figures for Figure 1 and Figure 1-supplement 2. As well, there are a number of errors in the figure legends and Materials and methods. For instance, in Figure legends 1 and supplement 2, the legends state that there are representative uCT images (although the English here is poor), but these are not provided. There are abbreviations that are not defined, e.g., U and TSS in Figure 3 legend, NC in Figures 5, 6 and 9. In the Materials and methods, the DXA procedure "of the head" does not make sense. The PTH treatment is given as 48 h, when they have time courses of 10-60 min. The entire manuscript needs careful review by the senior authors.

We apologize for these mistakes; the uCT images for Figure 1 are provided now and the abbreviations are provided as required in the figure legend.

DXA methodology was corrected to “Whole body composition exclusive of the head was performed using the PIXImus densitometer (GE-Lunar, Fairfield, CT, USA).”

PTH treatment in the methods section was corrected to “During osteogenic differentiation, PTH was administered by adding to osteogenic media every 48 hours for each group. For the Elisa assay, PTH was administered by adding to culture media for 10-60min.”

2. The Prrx1Cre/Zfp467 mice do not completely recapitulate the global knockout. They have reduced cortical thickness in the females, and no significant change in the males, opposite to the global knockouts. They say the global knockouts had increased cortical thickness, but it also was not significant, and the sexes were not given in that paper. Thus, they need to modify the language in the present manuscript. They also need to modify the conclusion of this section to "maybe attributable to changes in the MSC lineage".

We agree that Prx1Cre/Zfp467 did not completely recapitulate the global knockout. Although we observed reduced cortical thickness in females, the decreased thickness is subtle, but trabecular bone was much higher in Prx1Cre/Zfp467 mice (see Author response image 3). We agree that this terminology is confusing and likely outdated. We now refer to progenitor cells in the osteoblast lineage as skeletal stem cells.

3. The doses of PTH they have used with the cultured cells are very high, 5 x 10-8 M and 10-7 M. These doses often inhibit growth of osteoblastic cells. They never provide a dose response of PTH action, and there is no difference in effect between the two doses. In Figure 2., the decrease in Zfp467 mRNA expression is very transient, only observed at 10 min and returning to control by 30 min. They must provide protein levels of ZFP467 to establish that the gene expression is reflected in change in the protein, otherwise, it is not relevant.

Thank you for your suggestion. 5 x 10-8 M and 10-7 M PTH were also used in our previous studies (David E. Maridas, FASEB J. 2019; Yi Fan, Cell Metab. 2017) on MSCs, and we observed decreased adipogenesis and higher ostoegenesis; in addition no significant growth inhibition was observed. As reported by T. John Martin (Front Endocrinol, 2021), initial signaling in the PTH target cells through cAMP and protein kinase A (PKA) activation is extremely rapid (60 seconds). Quach et al. (J Biol Chem, 2011) also observed similar results that PTH suppressed zfp467 within 1 hour. Unfortunately, commercial ZFP467 antibody is not available, therefore it is not possible for us reflect the ZFP467 changes in protein level at this moment. We are trying to produce ZFP467 antibodies in rabbit for our future experiments.

4. In Figure 3, they should provide the relative levels of the three transcripts to each other, rather than using fold expression to the +/+ cells. Also, the luciferase data should be in units, not fold change to the NC (don't know what this is, "no transfection" or "no construct"?). Here, it is very clear that P1 is a poor promoter. The data here should also be tested as an ANOVA since they are using the NC to give fold change.

Thank you for your suggestions. NC represents for negative control. We added related information in the figure legend. Relative levels for three transcripts were presented instead of using fold change to +/+. Luciferase data were updated using units instead of fold change to NC.

5. Figure 4., the overexpression data should be judged by Westerns. The nuclear immunofluorescence expression of GATA1 and p50 should be quantitated.

Thank you for your suggestion, we added quantification for the immunofluorescence of GATA1 and p50.

6. Figure 4., supplement, it is not apparent whether the data are fold change to one of the samples, or relative expression. If the former, the sample seems to vary.

It is analyzed by fold change and based on 3 independent experiments.

7. Why don't they perform ChIP-seq for ZFP467? This would give them much more direct information as to how this transcription factor functions. They do not seem to have evidence that it binds to the promoters of the Pthr1 gene, but ChIP-seq may give them that.

We agree that ZFP467 did not bind to the promoter of Pth1r but via NFKB through non-canonical pathway activation. We plan to do CUT&TAG and combined with RNAseq data to find the direct downstream targets of ZFP467 in our future studies.

8. Figure 5C., they need to provide more information as to what each of these regions of p2 are, base pairs etc. Figure 5F., the PTHR1 data need to be quantitated.

Thank you for your questions. The detailed primer design for different regions of P2 were provided in Table 2, although base pairs would be too long to provide. PTHR1 data was quantitated.

9. Figure 6D. These immunoprecipitations are poor. There is no band in the input for RelB or NPAS1, and the quality of the blots is unsatisfactory.10. Figure 7A and B, the data should be presented as time course lines, not histograms. Panel C does not provide much information, except to say that there may be higher baseline cAMP in the -/- cells. They need to perform PTH treatment of the cells and determine if there is a difference in pCREB. Panels E-G are not particularly meaningful, and it is not apparent how they connect to the function of ZFP467.

Thank you for your suggestion. Since too many groups are involved, it would be hard to present the comparison among groups using time-course lines. As PTH was shown to enhance aerobic glycolysis, higher aerobic glycolysis in zfp467-/- cells compared to +/+ cells could indirectly reflect that Zfp467 -/- may have higher PTH1R and stronger PTH action.

11. Figure 8, they cannot call use of two high doses of PTH a dose response. In panel A, they can elute the alizarin red and obtain quantitation. What is 1 in panel D? In panel E, there is no PTH effect on Sp7, nor in the +/+ cells for Rankl (which is very surprising for COBs), and these exceptions need to be noted in the text.

Thank you for your suggestions. We added related absorbance quantification for ARS staining. We did find an almost 2-fold higher expression of Rankl in +/+ cells in response to PTH (Figure 8E), but not significant when compared to -/- cells. We agree that we did not observe any effect of PTH on Sp7.

12. The RNAseq of the two cell populations is disappointing since there do not appear to be any osteoblastic genes, but a number of chondrocyte genes changed. In addition, they note that the PI-3-K and MAPK pathways are differentially upregulated in the -/- cells, but these were never investigated.

Thank you for your questions. We agree that osteogenesis related genes were not observed in -/- cells, but we did find higher Rankl and Pth1r in -/- cells, which is in consistent with our mechanistic finding. Besides, several studies reported that SOX10 and NGFR are also involved in skeletal stem cell differentiation. We are now investigating the PI3K and MAPK pathways in our future study.

Reviewer #3 (Recommendations for the authors):First let me start by reporting that there is discrepancy in the figures depending on the version that is downloaded from the submission files. In the PDF version of the paper where the figures are incorporated in the text there is a graphical abstract (never mentioned in the text) and the rest of the Figures correspond to the text except for supplementary Figure 2 where some of the microCT (relative to cortical bone) data are missing. In the PDF version where the Figures are placed at the end of the document, there is no graphical abstract, Figure 1 is missing (it can only be seen in the other version and it is very small). In addition, there are 2 Figures relative to the phenotype of the AdipoCre Zfp467 mice. The second figure, which contains BMD and microCT (cortical data), is not mentioned in the Figure legend. Moreover, in Supplementary Figure 5, the panels are different depending on the version that is downloaded.

We apologize for these mistakes. The corrected Figure 1 and Figure 2 are provided now.

Labelling each figure would be very helpful as well as calling the Supplemental Figures only supplemental figures and not Figure x—figure supplemental.

Sorry about confusion surround the figure legends. However, the title for supplemental figures are made following *eLife* submission requirements.

1) The authors claim that PrrxCre Zfp467mice have an identical phenotype compared to Zfp467 global knock out mice (reference 14). However, Zfp467 global KO mice have increased cortical thickness (females) whereas in this paper the females have decreased cortical thickness. This discrepancy need to be explained and discussed. Moreover, the adipose tissue volume fraction is decreased in Zfp467 global knock out females but is not decreased in PrrxCre Zfp467 females. Overall, the trabecular phenotype is much less prominent and does not "perfectly recapitulate" the phenotype of the KO mice, therefore the authors cannot conclude that the changes in the Zfp467 KO mice are mostly attributable to the lack of Zfp467 in mesenchymal progenitors.

As noted, we agree that Prx1Cre/Zfp467 did not completely recapitulate the global knockout. Although we observed reduced cortical thickness in females, the decreased thickness is subtle, but higher trabecular bone was much higher in Prx1Cre/Zfp467 mice than controls. Differences could also be due to different measurement time points.

We agree that this terminology is confusing and likely outdated. We now refer to progenitor cells in the osteoblast lineage as skeletal stem cells.

2) In view of the differences in the phenotype between PrrxCre Zfp467 mice and global KO mice, to claim that Zfp467 KO mice have increased osteogenic differentiation and bone formation, histomorphometric data showing increased Osteoblast number or BFR should be reported.

Thank you for your suggestion, Histomorphometric data for Prx1Cre Zfp467 mice are now provided in Table 1**.**

3) Although the in vitro data are convincing and demonstrate that lack of Zfp467 increases the expression of the PTH1R and influences PTH-induced osteoblastogenesis, a clear proof of this mechanism would be treatment of PrrxCre Zfp467 and control mice with PTH to assess the difference in the magnitude of the response to PTH in vivo.

Thank you for your suggestions, we decided to treat Zfp467 global ko mice with PTH for 1 week and compared the response to control mice. Although we did not find significant differences between Vehicle and PTH group in +/+ mice, PTH treated -/- mice had much higher BT/TV, Conn.D and lower Tb.Sp, SMI than PTH treated +/+ mice. See Figure 8F.